# BEYOND OUTLIERS:
# A STUDY OF OPTIMIZERS UNDER QUANTIZATION

**Georgios Vlassis**[*]
ETH Zurich
gvlassis@ethz.ch

**Saleh Ashkboos**[*]
ETH Zurich
saleh.ashkboos@inf.ethz.ch

**Alexandra Volkova**
ISTA
avolkova@ist.ac.at

**Torsten Hoefler**
ETH Zurich
htor@ethz.ch

**Dan Alistarh**
ISTA & Red Hat AI
dan.alistarh@ist.ac.at

## ABSTRACT

As new optimizers gain traction and model quantization becomes standard for efficient deployment, a key question arises: *how does the choice of optimizer affect model performance in the presence of quantization?* Despite progress in both areas, systematic evidence on optimizer–quantization interactions remains limited. To fill this gap, we study the impact of optimizer choice on model robustness under quantization, considering both post-training quantization (PTQ), and quantization-aware training (QAT). We first train full-precision models, ranging from 50M to 1.5B parameters, with six optimizers, to explore the hyperparameter landscape, and establish well-tuned baselines. We then apply PTQ to evaluate how model performance degrades when trained with different optimizers. We find that outlier-related metrics, such as the *max-to-mean* ratio (MMR) and Kurtosis, fail to predict the PTQ performance across different optimizers. We show analytically that this is due to the MMR capturing only isolated layer errors, while ignoring how quantization errors accumulate and propagate through the network. To study the QAT degradation, we train quantized models from scratch and compare them to our original-precision baselines. We find that optimizers performing well in the original pretraining setup may not remain optimal under QAT, and that models trained with Shampoo show the lowest accuracy degradation. Finally, we derive scaling laws for quantization-aware training under different optimizers, showing that Shampoo achieves the highest *parameter efficiency* of all tested optimizers.

## 1 INTRODUCTION

Large language models (LLMs) based on the Generative Pretrained Transformer (GPT) architecture have billions of parameters, and this number continues to grow, making both training and deployment increasingly challenging. Quantization is a common technique to make the training and deployment of large-scale neural networks more efficient. In general, LLM quantization can be categorized into two types: **Post-Training Quantization** (PTQ), where a model is trained in full precision and then quantized, and **Quantization-Aware Training** (QAT), where quantization is incorporated directly during training. The effectiveness of both methods ultimately depends on how well models can maintain accuracy in the face of quantization-induced degradation.

Post-training quantization (PTQ) is one of the most important techniques to address the challenges of LLM deployment. To accelerate the inference, both weights and inputs (also known as "activation") are quantized, enabling most computations to be performed in low precision. This technique is known as joint quantization (Ashkboos et al., 2024; 2023; Xiao et al., 2023; Liu et al., 2024). Several studies (Dettmers et al., 2022; Wei et al., 2022; Bondarenko et al., 2023) have shown that joint quantization is challenging due to the presence of outlier features (or OF) in the input matrices. To mitigate OF, various approaches have been proposed, such as reducing kurtosis (Nrusimha et al.,

---
[*]Equal contribution.

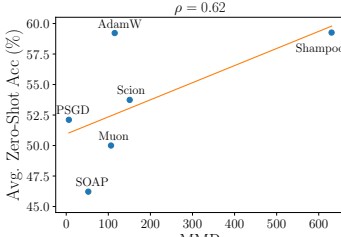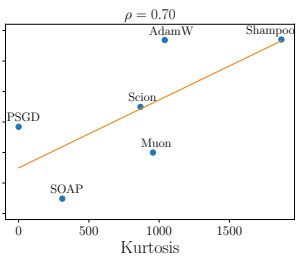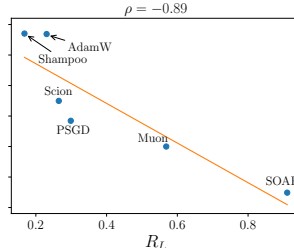

Figure 1: Accuracy correlation with different metrics for the 760M model. Traditional outlier-sensitive metrics like MMR (**Left**) and kurtosis (**Center**) show little to no correlation (measured by $\rho$) with model accuracy, whereas our proposed metric (**Right**) correlates strongly with the model's zero-shot performance. MMR and kurtosis are computed row-wise, on the output of the last transformer block.

2024; Akhondzadeh et al., 2025) or the Max-Median Ratio (MMR) (He et al., 2024) prior to PTQ, often through techniques like rotations (Ashkboos et al., 2024) or architectural modifications (He et al., 2024). However, it remains unclear how the optimization process during pretraining affects the presence of such outlier features in the resulting model.

Quantization-aware training (QAT) is another important technique for reducing the deployment cost of large language models (LLMs). Unlike post-training quantization, QAT applies quantization during training, allowing the model to adapt to low-precision computations (Panferov et al., 2025; Ashkboos et al., 2025; Xi et al., 2024). Typically, the forward pass is quantized while the backward pass remains in high precision, with the gradient of the rounding function estimated using the Straight-Through Estimator (Bengio et al., 2013). Most existing works adopt the same optimization process as in full-precision training, and AdamW remains the most commonly used optimizer. Yet, the impact of using different optimizers on the final performance of QAT remains unclear.

Over the last decade, Adam (Kingma & Ba, 2015) and AdamW (Loshchilov & Hutter, 2017) have become the default optimizers for LLM training due to their simplicity and stability. At the same time, several alternative optimizers, such as PSGD (Li, 2015), Shampoo (Gupta et al., 2018), Muon (Jordan, 2024), SOAP (Vyas et al., 2025), and Scion (Pethick et al., 2025), have been proposed. Recently, concurrent studies (Wen et al., 2025; Semenov et al., 2025) have benchmarked optimizers in practice by training models with different hyperparameter settings to evaluate their empirical benefits. These studies are restricted to high-precision models, and despite advances in both optimization and quantization techniques, their interaction remains underexplored.

In this work, we investigate the gap between optimization and quantization of LLMs for efficient deployment. For post-training quantization (PTQ), we investigate the question: "**Do two models with the same validation loss but trained with different optimizers perform similarly under PTQ?**" To answer this, we train the same model to the same validation loss using different optimizers and compare the accuracy drop after quantization. We then evaluate the impact of different optimizers during quantization-aware training (QAT), asking: "**How sensitive is QAT to the choice of optimizer—does the optimizer that performs best in full precision maintain its advantage under QAT?**" Finally, we explore "**How well do our findings transfer to larger models?**" and provide a scaling law for QAT performance across different optimizers. Our contributions are as follows:

1. We present the first systematic study of the interaction between optimization and quantization, by training models ranging from 50M to 1.5B parameters with six different optimizers, and evaluating full-precision training (FP), post-training quantization (PTQ), and quantization-aware training (QAT) schemes.

2. For FP training, we sweep over different hyperparameters for each optimizer–model combination, and train each using the Chinchilla-optimal data-to-model ratio. We find that Muon outperforms other optimizers across nearly all model sizes.

3. For PTQ, we find that full-precision accuracy does not correlate with quantized recovery, nor with standard metrics of outlier quantification, challenging current understanding. Instead, we provide a first theoretical analysis of error propagation during quantization, and

propose a new metric for predicting the accuracy of quantized models. In particular, we demonstrate that models trained with Shampoo are more robust to quantization.

4. For QAT, we again observe that optimizers performing best in full precision are not necessarily optimal under QAT. Among all optimizers, the model trained with Shampoo yields the lowest accuracy degradation compared to its high-precision counterpart in almost all of the model sizes. We further provide a scaling law for different optimizers under 4-bit QAT and show that Shampoo achieves the highest parameter efficiency, supporting the transferability of our results to larger scales.

## 2 BACKGROUND AND EXPERIMENTAL SETUP

In this section, we first present the model architecture used in our experiments and the optimizers applied during training. We then describe the hyperparameter tuning strategy employed in our experiments. Finally, we provide the training and evaluation details for both PTQ and QAT.

**Model Architecture.** We use the OLMo2 architecture (OLMo et al., 2024), which integrates several recent architectural improvements. In particular, OLMo2 combines no biases, rotary positional embeddings (Su et al., 2024), RMSNorm (Zhang & Sennrich, 2019), reordered pre-normalization (Liu et al., 2022; OLMo et al., 2024) and QKNorm (Henry et al., 2020). Our only modifications on top of OLMo2 are the tying of the input and output weights, and the use of the ReLU$^2$ activation function (So et al., 2022). We train models with 50M, 125M, 350M, 500M, 760M and 1.5B parameters, and we list the exact architectural details in Table 1.

**Optimizers.** We evaluate six optimizers. **AdamW** serves as the standard baseline for training LLMs. To capture curvature-aware methods, we include **PSGD** and **Shampoo**. In addition, we also consider **Muon** and **Scion**, as methods with motivations rooted in feature learning (Yang et al., 2023). Lastly, we include **SOAP**, which operates by rotating gradients from and to a different eigenspace. We provide a complete analysis of the computational complexity and memory overhead of each optimizer for a linear layer, with details provided in Appendix A.5.

**Experimental Setup and Hyper-parameter Protocol.** As a full-precision baseline, we train models in BFloat16 with the following protocol for hyper-parameter selection:

1. **Optimizer Parameters.** We initialize each optimizer with hyperparameters recommended in its original paper and perform sequential one-dimensional sweeps to determine the optimal settings. These sweeps are conducted on the smallest 50M model, and the resulting configurations are applied across all experiments.

2. **Learning Rate Selection.** After fixing the optimizer hyperparameters, we sweep over eight different learning rate values and fully train each model–optimizer pair to find the optimal learning rate. For the 1.5B model, we select the optimal learning rate for each optimizer based on results from the 760M model, and then test four additional values (two higher and two lower) to identify the best learning rate. We train our 1.5B model using AdamW (as the baseline), Muon (the best-performing optimizer without quantization), and Shampoo (the most effective optimizer for preserving accuracy under quantization).

**Datasets, Training Steps, and Evaluation Taks.** We train all models on the ClimbMix dataset (Diao et al., 2025), a high-quality mixture of 400B tokens, following the Chinchilla training regime. For each model size, we define the Common Loss (or **CL**) as the lowest evaluation loss achieved by all optimizers using their best hyperparameters with at most a 20x token-to-parameter ratio. This CL checkpoint is then used for applying the PTQ scheme. For each model–optimizer pair, we select the best hyperparameters from high-precision experiments and perform quantized training with a fixed 20x token-to-parameter ratio. We evaluate our models on three standard zero-shot benchmarks that offer early signal: PIQA (Bisk et al., 2020), HellaSwag (Zellers et al., 2019), and ARC-Easy (Clark et al., 2018). We report the average accuracy across these tasks. All evaluations are performed using the LM Evaluation Harness (Gao et al., 2021) with default parameter settings. We always use 8196 samples from the test set of ClimbMix dataset to calculate the statistics, each with 1024 tokens.

| Model | Exact Num. Parameters | Blocks ($L$) | $Q$ heads | $KV$ heads | Width ($d$) | Toks | Iters. | CL |
|-------|----------------------|--------------|-----------|------------|-------------|------|--------|------|
| 50M | 49,748,760 | 4 | 6 | 2 | 768 | 1B | 2,000 | 3.73 |
| 125M | 121,813,686 | 6 | 9 | 3 | 1152 | 2B | 4,000 | 3.32 |
| 350M | 325,335,892 | 11 | 12 | 4 | 1536 | 6B | 12,000 | 2.97 |
| 500M | 477,024,972 | 17 | 12 | 4 | 1536 | 10B | 20,000 | 2.80 |
| 760M | 729,974,655 | 17 | 15 | 5 | 1920 | 14B | 28,000 | 2.75 |
| 1.5B | 1,489,423,554 | 25 | 18 | 6 | 2304 | 30B | 60,000 | 2.57 |

Table 1: Training configurations used for our experiments. **CL** denotes the common loss, which is the lowest validation loss that all optimizers can achieve for a given model size and number of tokens (used for the PTQ experiments).

**Quantization.** We apply the *round-to-nearest-integer* scheme for PTQ experiments. For each tensor, every row is normalized by its largest magnitude value and then scaled by the largest representable number, a method known as *symmetric AbsMax row-wise quantization*. We use 4-bit quantization for both the weights and inputs of all linear layers, while keeping the remaining modules in their original precision. For QAT, we train our models using QuEST (Panferov et al., 2025), a state-of-the-art stable quantization-aware training scheme for very low precisions (e.g., 4-bit). In the forward pass, QuEST applies the Hadamard transform to both the inputs and weights of linear layers, followed by the selection of an optimal clipping ratio to minimize MSE before quantization. During the backward pass, it masks gradients that deviate significantly from the clipping value. We keep the backward pass (and non-linear modules) in the original precision following standard QAT.

## 3 RESULTS AND ANALYSIS

In this section, we present our main results, analyses, and findings from the experiments. We begin with full-precision training, which serves as the baseline for our study and show that the outlier-related metric MMR correlates with learning rates across different optimizers. Next, we report the results of post-training quantization (PTQ) and provide a theoretical analysis of error propagation using our ABC decomposition framework. Finally, we present the results of our quantization-aware training (QAT) experiments, along with the corresponding scaling laws.

### 3.1 FULL-PRECISION

Table 2 shows the zero-shot accuracies of our (unquantized) models across different optimized learning-rates. Except for the 50M and 125M models, Muon consistently outperforms other optimizers in zero-shot accuracy, aligning with the concurrent study of Wen et al. (2025). The performance gap widens for larger models (from 0.01% in the 350M to 1.03% in 1.5B model), indicating that **Muon is the best optimizer in high precision, when hyperparameters are tuned**. We provide additional results in Appendix A.1.

| Optimizer | Model Size | | | | | |
|-----------|-------|-------|-------|-------|-------|-------|
| | 50M | 125M | 350M | 500M | 760M | 1.5B |
| AdamW | 43.75 | 48.64 | 56.58 | 60.39 | 63.90 | 67.93 |
| Muon | 45.03 | 49.62 | **58.08** | **61.86** | **64.63** | **69.19** |
| PSGD | 45.08 | 49.95 | 55.99 | 60.85 | 61.47 | N/A |
| Scion | 45.41 | 49.83 | 57.53 | 61.68 | 63.82 | N/A |
| Shampoo | 44.81 | 49.53 | 56.51 | 61.03 | 63.05 | 68.16 |
| SOAP | **45.73** | **50.24** | 58.07 | 61.34 | 62.27 | N/A |

Table 2: Average zero-shot accuracy ($\uparrow$) for the full-precision models, trained with BFloat16. Results for the test loss, which show similar trends, can be found in Table 5.

We also investigate how varying the learning rate impacts both the validation loss and the MMR, which is used to study the outlier patterns, in Figure 2. Our analysis reveals a clear trend: **increasing the learning rate consistently leads to higher MMR values, independent of the optimizer used**. This suggests that larger learning rates tend to amplify the relative difference between the maximum and median value in the input tensor, potentially indicating larger outliers. Furthermore, across all optimizers in our experiments, **Muon consistently achieves the lowest MMR**. We provide results for other models in Appendix A.1.

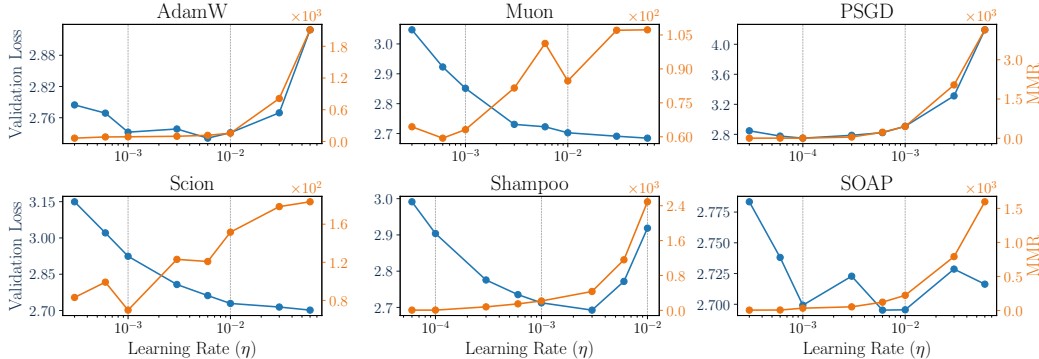

Figure 2: The effect of changing learning-rate $\eta$ on the validation loss and MMR in different optimizers for 760M model. We average the MMR over the rows in the input tensor of the last linear layer (before head).

## 3.2 POST-TRAINING QUANTIZATION

In the previous section, we showed that models trained with different optimizers yield substantially different MMR values. We would thus expect models with higher MMR (larger outlier features) to exhibit more severe degradation than networks with lower MMR. Here, we examine to what extent this intuition holds in practice. To facilitate comparison, we train all models to a common loss (CL), defined as the lowest validation loss achievable by all optimizers, given the same number of tokens. Since all the networks were trained to the same CL, the downstream performance before quantization is very similar. Hence, instead of measuring the performance degradation, we can equivalently look at the final downstream performance.

According to our results, presented in Table 3, the optimizer that consistently leads to the least degraded networks, at least for model sizes above 125M, is Shampoo. This is counter-intuitive, since Shampoo's networks are also characterized by the highest MMR. Following folklore knowledge, this would imply that it would deteriorate the most under quantization. Also noteworthy is the fact that optimizers with low MMR (e.g. Muon), that would be expected to perform near the top, show a significant *decline* in accuracy. Plotting the row-wise MMR of the output of the last transformer block against the average downstream accuracy after PTQ, as in Figure 1-left, reveals that the two metrics are uncorrelated. Changing the metric we use to quantify outliers from MMR to kurtosis, as in Figure 1-center, does not change the pattern.

To understand this discrepancy, we introduce a new metric, which not only correlates strongly with the PTQ performance, but also allows us to track the quantization error through the network, as well as decompose it into independent and interpretable terms.

**Theoretical analysis.** We start with a brief description of the setup for which our analysis works, and present a concise summary of the main results. A full derivation can be found in Appendix A.4.

We consider a Feed-Forward Network (FFN) with $L$ modules $f_\ell(\cdot)$, for $\ell \in \{1, \dots, L\}$. We denote the activations of the network with $h_\ell \in \mathbb{R}^{n_\ell}$, for which $h_\ell = f_\ell(h_{\ell-1})$, with

| Optimizer | Model Size | | | | | |
|---|---|---|---|---|---|---|
| | 50M | 125M | 350M | 500M | 760M | 1.5B |
| AdamW | 40.23 | 45.15 | 49.23 | 55.17 | 59.22 | 62.51 |
| Muon | 41.88 | 45.23 | 47.42 | 50.60 | 50.00 | 47.75 |
| PSGD | 42.69 | **47.39** | 50.09 | 54.01 | 52.11 | N/A |
| Scion | 42.44 | 46.04 | 49.80 | 52.14 | 53.74 | N/A |
| Shampoo | 40.84 | 45.68 | **53.93** | **55.65** | **59.26** | **63.88** |
| SOAP | **43.36** | 46.35 | 49.08 | 50.91 | 46.22 | N/A |

Table 3: Average zero-shot accuracy ($\uparrow$) after we applied PTQ (row-wise W4A4) to the networks with the same common-loss (CL). The pre-quantization models used here have the same validation loss (shown in Table 1). Thus, the post-quantization performance is indicative of the degradation due to quantization. The equivalent table for test loss is in Appendix A.2. GPTQ (Table 7) and Llama 2 (Table 8) have similar results.

$h_0$ being the input of the network. Notably, this construction allows for any arbitrary transformation $f_\ell(\cdot)$ (e.g. linear layers, convolutions, self-attention etc.), at any level of granularity.

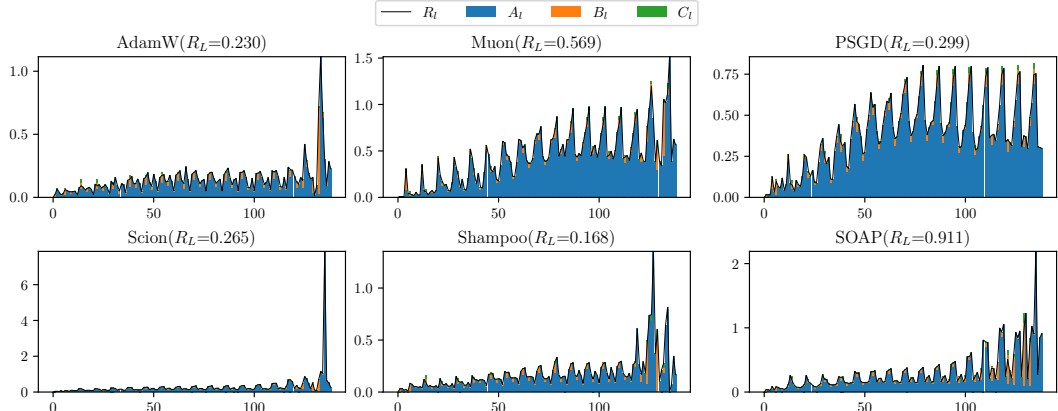

Figure 3: ABC decomposition for the 760M models. The x-axis shows the module index $\ell$. To use the stacked bar plot, and because $C_\ell$ can be negative, we actually visualize $|C_\ell|$. If $R_\ell$, which should track $A_\ell + B_\ell + C_\ell$, is below the green bar, it means $C_\ell$ is negative, and the green bar only shows the magnitude. In the majority of cases, the biggest component is $A_\ell$ (accumulated error), followed by $B_\ell$ (layerwise error), followed by $C_\ell$ (error interaction). The regular dips in the plots correspond to RMSNorm layers, which attenuate the quantization error instead of amplifying it.

Now assume that we quantize the modules of the network. We can choose to quantize part or all of the modules, with any arbitrary PTQ scheme. If we denote the quantized modules with $f_\ell^{\mathrm{q}}$, the quantized activations become $\boldsymbol{h}_\ell^{\mathrm{q}} = f_\ell^{\mathrm{q}}(\boldsymbol{h}_{\ell-1}^{\mathrm{q}})$. Thus, there is both a change in the input, from $\boldsymbol{h}_{\ell-1}$ to $\boldsymbol{h}_{\ell-1}^{\mathrm{q}}$, as well as a change in the function, from $f_\ell(\cdot)$ to $f_\ell^{\mathrm{q}}(\cdot)$. The change in the input $\boldsymbol{h}_{\ell-1}$ comes from the propagation of the quantization error through the previous $(\ell-1)$ modules, while the change in the function $f_\ell(\cdot)$ is the newly introduced layerwise perturbation.

We aim to precisely quantify these changes. To do so, we study the difference in the activations $\boldsymbol{\Delta h}_\ell := \boldsymbol{h}_\ell^{\mathrm{q}} - \boldsymbol{h}_\ell$. We prove (Appendix A.4) that this difference can be written as $\boldsymbol{\Delta h}_\ell = \boldsymbol{a}_\ell + \boldsymbol{b}_\ell$, with:

$$\boldsymbol{a}_\ell := \frac{\overbrace{(f_\ell^{\mathrm{q}}(\boldsymbol{h}_{\ell-1}^{\mathrm{q}}) - f_\ell^{\mathrm{q}}(\boldsymbol{h}_{\ell-1}))}^{\text{Change in the input under } f_\ell^{\mathrm{q}}(\cdot)} + \overbrace{(f_\ell(\boldsymbol{h}_{\ell-1}^{\mathrm{q}}) - f_\ell(\boldsymbol{h}_{\ell-1}))}^{\text{Change in the input under } f_\ell(\cdot)}}{2},$$

$$\boldsymbol{b}_\ell := \frac{\overbrace{(f_\ell^{\mathrm{q}}(\boldsymbol{h}_{\ell-1}^{\mathrm{q}}) - f_\ell(\boldsymbol{h}_{\ell-1}^{\mathrm{q}}))}^{\text{Change in the function under } \boldsymbol{h}_{\ell-1}^{\mathrm{q}}} + \overbrace{(f_\ell^{\mathrm{q}}(\boldsymbol{h}_{\ell-1})) - f_\ell(\boldsymbol{h}_{\ell-1}))}^{\text{Change in the function under } \boldsymbol{h}_{\ell-1}}}{2}.$$

From the definitions, we see that $\boldsymbol{a}_\ell$ is the average of the change in the input under $f_\ell^{\mathrm{q}}(\cdot)$, and the change in the input under $f_\ell(\cdot)$. On the other side, $\boldsymbol{b}_\ell$ is the average of the change in the function under $\boldsymbol{h}_{\ell-1}^{\mathrm{q}}$, and the change in the function under $\boldsymbol{h}_{\ell-1}$. Hence, $\boldsymbol{a}_\ell$ captures the effect of the change in the input, while $\boldsymbol{b}_\ell$ encodes the change in the function.

In order to reduce $\boldsymbol{\Delta h}_\ell$ to an interpretable number, we compute the $L_2$-norm. Since $\|\boldsymbol{\Delta h}_\ell\|$ should be interpreted relative to the scale of the initial unquantized activations, we also normalize by $\|\boldsymbol{h}_\ell\|$. The norm of the relative change in the inputs is then:

$$r_\ell := \frac{\|\boldsymbol{\Delta h}_\ell\|}{\|\boldsymbol{h}_\ell\|} = \frac{\|\boldsymbol{a}_\ell + \boldsymbol{b}_\ell\|}{\|\boldsymbol{h}_\ell\|}.$$

To be able to use the Law of Cosines (which we use to prove the following decomposition), we instead study the square of this quantity $R_\ell := r_\ell^2$. Intuitively, this quantity tracks the deviation of the quantized network from the original one across its layers. Hence, we would expect $R_L$ at the output of the network[1] to correlate with loss degradation.

---

[1] Calculating $R_L$ requires four forward passes (one for each of $f_L(\boldsymbol{h}_{L-1})$, $f_L(\boldsymbol{h}_{L-1}^{\mathrm{q}})$, $f_L^{\mathrm{q}}(\boldsymbol{h}_{L-1})$, $f_L^{\mathrm{q}}(\boldsymbol{h}_{L-1}^{\mathrm{q}})$), and is thus more computationally demanding than MMR. However, it is also much more pre-

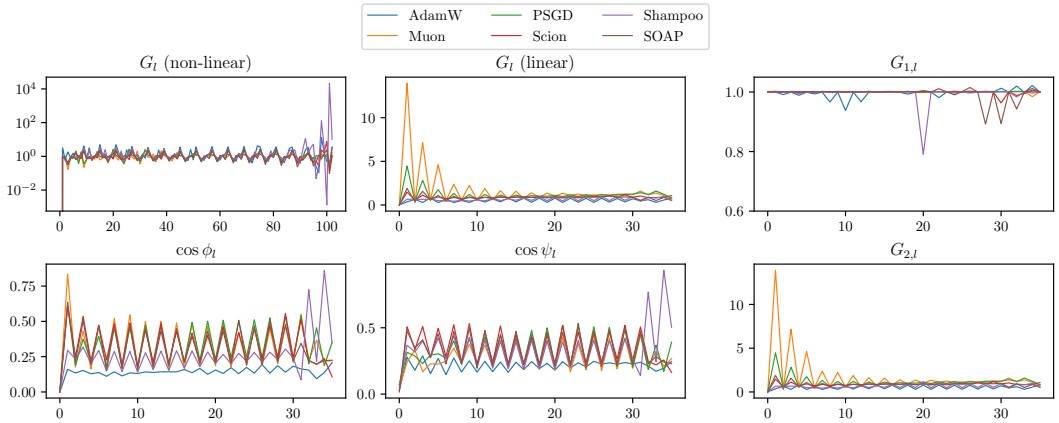

Figure 4: Gain decomposition for the 760M models with different optimizers. The x-axis shows the module index $\ell$. Reassuringly, the dips coincide with those of Figure 3.

For any module of the network, the square of the norm of the relative change in the activations $R_\ell$ can be decomposed *exactly* into three terms (see Appendix A.4):

$$R_\ell = A_\ell + B_\ell + C_\ell$$

$$A_\ell := \left(\frac{\|\boldsymbol{a}_\ell\|}{\|\boldsymbol{h}_\ell\|}\right)^2, \ B_\ell := \left(\frac{\|\boldsymbol{b}_\ell\|}{\|\boldsymbol{h}_\ell\|}\right)^2, \ C_\ell := 2\frac{\langle \boldsymbol{a}_\ell, \boldsymbol{b}_\ell\rangle}{\|\boldsymbol{h}_\ell\|^2}$$

Here $A_\ell$ reflects the quantization error accumulated from the previous $(\ell-1)$ modules, $B_\ell$ expresses the quantization error introduced in the $\ell$-th layer, and $C_\ell$ captures the interaction between the two.

To understand how a module transforms the quantization error of the previous layer from $R_{\ell-1}$ to $A_\ell$, we can define the *gain*:

$$G_\ell := \frac{A_\ell}{R_{\ell-1}}.$$

In general, without assumptions about the functional form of the module, we cannot further analyze $G_\ell$ (even though we can use $A_\ell$ and $R_{\ell-1}$ to calculate it). However, for a linear layer under weight and activation quantization with $\boldsymbol{h}_\ell^{\mathrm{q}} = (\boldsymbol{W}_\ell + \boldsymbol{\varepsilon}_\ell^W)\boldsymbol{h}_{\ell-1}^{\mathrm{q}} + \boldsymbol{\varepsilon}_\ell^h$, where $\boldsymbol{\varepsilon}_\ell^W$ and $\boldsymbol{\varepsilon}_\ell^h$ are quantization noise introduced by weight and activation quantization respectively, there is a very intuitive decomposition of $G_\ell$. Specifically, we can write:

$$G_\ell = G_{1,\ell}G_{2,\ell}, \ G_{1,\ell} := \left(\frac{\|\boldsymbol{W}_\ell + \frac{1}{2}\boldsymbol{\varepsilon}_\ell^W\|_*}{\|\boldsymbol{W}_\ell\|_*}\right)^2, \ G_{2,\ell} := \left(\frac{\cos\phi_\ell}{\cos\psi_\ell}\right)^2$$

where $\phi_\ell$ is the angle[2] between the change in activations $\boldsymbol{\Delta h}_{\ell-1}$ and the weight matrix after quantization, and $\psi_\ell$ is the angle between the original activations and the unquantized weights.

The term $G_{1,\ell}$, which we call the "spectral ratio", reflects the change in the spectral norm of the weights due to quantization. We call the term $G_{2,\ell}$ the "alignment ratio", since $\cos\phi_\ell \in [0,1]$ is the alignment between the change in the activations and the weights after quantization, while $\cos\psi_\ell \in [0,1]$ is the alignment between the incoming activations and the original weights.

**Empirical Validation.** To validate our analysis, we monitor $R_\ell$ along the residual path of the transformer. We analyze the models at their most granular level, treating layer normalizations, residual connections, linear layers and activation functions separately, with the exception of the multi-head self-attention (MHSA) module, which we consider as a single unit[3]. In our framework,

---

dictive of quantization degradation (see Figure 1). In our code, we include an optimized implementation which closely follows the mathematical derivation of $R_l$.

[2]Here, we define the angle between a matrix $A$ and a vector $x$ as $\arccos\left(\frac{\|Ax\|}{\|A\|_*\|x\|}\right)$. This is a non-standard definition, that we nonetheless find useful to quantify how close $x$ is to maximizing $\|Ax\|$. Notably, this angle lies between 0 and $\frac{\pi}{2}$, in contrast to angles between vectors, which lie between 0 and $\pi$

[3]We do this simply because MHSA internally has three branches, keys, queries and values, making it non-obvious what is considered the residual path.

the input $h_0 \in \mathbb{R}^{n_0}$ was assumed to be a vector. Because transformers process multiple vectors (corresponding to tokens), we need to use a summary statistic. As long as the statistic is linear, the ABC decomposition still holds with $\text{Stat}(R_\ell) = \text{Stat}(A_\ell) + \text{Stat}(B_\ell) + \text{Stat}(C_\ell)$. Unless otherwise noted, we use the average as our summary statistic [4].

First, we confirm our intuition that $R_L$ should predict the degradation due to quantization in Figure 1(right). This suggests that $R_\ell$ will be an informative quantity about how the quantization error propagates through the network.

Figure 3 shows the ABC decomposition for the 760M models. Here, we make two observations. First, in nearly all the cases, $R_\ell$ is dominated by $A_\ell$, while $B_\ell$ and $C_\ell$ play a minor role. That is, **even if MMR is a good predictor of the quantization error introduced by quantizing a specific layer, the total quantization error**, represented by $R_\ell$, **is mostly determined by the amplified error from the previous module**, represented by $A_\ell$. Second, **networks trained with different optimizers exhibit very different error propagation behaviors**. Even though all the networks exhibit oscillatory patterns with the $R_\ell$ generally increasing with depth, some optimizers (e.g. AdamW, Scion) lead to networks with prominent spikes toward the end, while others (e.g. PSGD, Muon) have a "flatter" profile.

Finally, we measure the gain and its related quantities in Figure 4. We separately show the gain of non-linear modules, and the gain of the linear layers, which we can analyze to $G_{1,\ell}$ and $G_{2,\ell}$. In addition, we show the alignment factors $\cos\phi_\ell$ and $\cos\psi_\ell$. We notice that, in the non-linear modules, different optimizers produce very similar $G_\ell$ profiles, with the exception of Shampoo, which exhibits not only two upward spikes but also a sharp downward drop. In contrast, the patterns observed for linear layers are more distinguishable for different optimizers. Notably, Muon has the highest gain for linear layers, possibly explaining its sharp quality degradation despite that it has relatively low MMR. On the other hand, AdamW, along with Shampoo, have the lowest gains.

As for the decomposition of the gain for linear layers, we see that the spectral ratio is close to 1 for all optimizers. That is, the spectral norm of the quantized weights is not significantly different than that of the original weights. Hence, $G_\ell$ is almost entirely determined by $G_{2,\ell}$. Looking at $\cos\phi_\ell$, **we see that for Shampoo, and even more so for AdamW, the change in the activations is less aligned with the quantized weights**. The other optimizers mostly follow the same trends. Lastly, the factor $\cos\psi_\ell$ is similar across most optimizers with the exception of AdamW, for which the input activations are even less aligned with the weights.

### 3.3 Quantization-Aware Training and Scaling Law

Finally, we study quantization-aware training (QAT), where we empirically investigate the extent of performance degradation under QAT compared to full-precision training, and how strongly this degradation depends on the choice of optimizer. We then provide a scaling law for QAT experiments to examine the transferability of our results to larger-scale models.

**Empirical Results.** We use the state-of-the-art 4-bit QAT QuEST scheme (Panferov et al., 2025), where weights and activations are quantized to 4-bits on the forward pass using row-wise symmetric quantization. We train each model–optimizer pair under the same compute budget, following the Chinchilla-optimal ratio (Hoffmann et al., 2022), using the best full-precision (FP) hyperparameters. We report the QAT evaluation losses across model sizes and optimizers in Table 9, as well as averaged zero-shot accuracy results (together with relative differences to their FP-baseline) in Table 4. First, we see that the ranking of the optimizers differs from the full-precision regime in almost all of the model sizes: no single best-performing optimizer could be identified for quantization-aware training. However, **Shampoo remains the most effective optimizer in minimizing accuracy degradation during quantization-aware training.** Optimizers differ in how much accuracy they lose compared to their full-precision baseline, which is why promising results of an optimizer in FP do not always lead to its strong performance under QAT. Overall, we find that **full-precision results do not reliably predict QAT behavior**.

---

[4]Figure 3 is the only exception where we use the truncated average (average after we ignore the top 1% of the values) as our summary statistic. $\text{Trunc}(R_\ell)$ and $\text{Avg}(R_\ell)$ only differ non-negligibly for a single intermediate layer for Shampoo, where a spike in $\text{Avg}(R_\ell)$ would force us to use log-scale.

| Optimizer | Model Size | | | | | |
|---|---|---|---|---|---|---|
| | 50M | 125M | 350M | 500M | 760M | 1.5B |
| AdamW | 43.37 (−**0.87**) | 48.08 (−**1.15**) | 54.64 (−3.43) | 60.07 (−0.53) | 62.22 (−2.63) | 66.82 (−1.63) |
| Muon | 44.07 (−2.13) | 48.33 (−2.60) | 55.19 (−4.98) | **61.05** (−0.73) | 62.32 (−3.57) | 67.08 (−2.11) |
| PSGD | 44.33 (−1.66) | 48.82 (−2.26) | 51.25 (−8.47) | 60.32 (−0.87) | 60.50 (−1.58) | N/A |
| Scion | **44.86** (−1.21) | 48.20 (−3.27) | 56.01 (−2.64) | 60.72 (−1.56) | 62.30 (−2.38) | N/A |
| Shampoo | 43.80 (−2.23) | 48.31 (−2.46) | 55.02 (−2.64) | 60.80 (−**0.38**) | **62.76** (−**0.46**) | **67.34** (−**1.20**) |
| SOAP | 44.41 (−2.89) | **49.38** (−1.71) | **56.61** (−**2.51**) | 60.58 (−1.24) | 61.79 (−0.77) | N/A |

Table 4: Average zero-shot accuracy (↑) for the models trained with 4-bit QAT via QuEST. We report the difference in accuracy relative to the full-precision baseline in the brackets. We **bold** the best accuracy and the smallest degradation for each model size. The corresponding results for test loss are presented in Table 9. Results for a smaller data-to-model ratio can be found in Table 10.

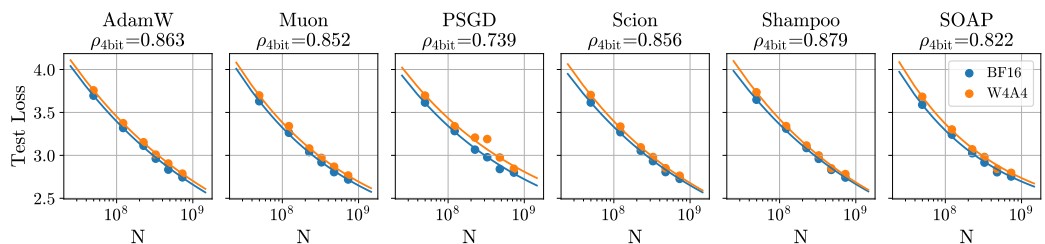

Figure 5: Scaling Laws for each optimizer, for full precision (BF16) and QAT (W4A4). For each optimizer, we report the parameter efficiency $\rho$ of 4-bit QAT in the subplot title. Shampoo has the highest parameter efficiency, followed by AdamW.

**Scaling Laws.** To provide a comprehensive comparison of QAT performance, we employ scaling laws. For each optimizer, the scaling law takes the form of Equation (1) from (Hoffmann et al., 2022), where $L$ is final test loss (cross-entropy), $N$ is model parameter count and $D$ is number of training tokens:

$$L = \frac{A}{(N \cdot \rho)^\alpha} + \frac{B}{D^\beta} + E. \tag{1}$$

We fit distinct sets of hyperparameters for each optimizer. To account for the QAT results we use the per-optimizer parameter efficiency $\rho$, introduced by (Kumar et al., 2024), as an extra hyperparameter in the law. By convention, we set $\rho = 1$ for full-precision training, and we denote $\rho_{4bit}$ the parameter efficiency in QAT W4A4 regime. Intuitively, an optimizer with higher $\rho$ has higher parameter efficiency under QAT. For example, a model of size $N$ trained in 4 bits QAT performs equivalently to the model of size $\rho_{4bit} \cdot N$ trained in full-precision.

Since our experiments use fixed $\frac{D}{N} = 20$, the full form of the law would be under-determined, and we report the coefficients of a *iso-compute* law:

$$L = \frac{A'}{(N \cdot \rho)^\alpha} + E. \tag{2}$$

Here, the parameters $A'$ and $\alpha$ describe the scale and the curvature of the power law, and $E$ corresponds to the irreducible error. We estimate parameters of the law with robust non-linear Huber loss ($\delta = 10^{-3}$) on $(\log N, \log D, \log L)$ manifold, and report the values with confidence intervals for each optimizer, obtained by leave-one-out cross-validation, in Appendix A.6.

Figure 5 shows our fitted curves for different optimizers. The single scalar $\rho_{4bit}$ serves a quantitative measure of robustness to quantization: a higher value means the quantized model retains a larger effective parameter capacity. Similar to our QAT results, we can see that among the optimizers evaluated in this study, **Shampoo yields the highest $\rho_{4bit}$, indicating that it is the most resilient to quantization** and preserves model performance more effectively. Also, the top optimizers in terms of parameter efficiency (Shampoo, AdamW, Scion) are also leading PTQ recovery (Table 3).

## 4 CONCLUSION AND FUTURE WORK

In this paper, we present a systematic study of the role of optimizer choice in the accuracy drop during quantization. First, we establish a full-precision baseline by training a single model with multiple optimizers in the original precision, tuned over eight different learning rates and optimized hyperparameters. We then apply post-training quantization, showing that existing outlier-centric metrics like MMR or Kurtosis, used in previous works, are not predictive of PTQ accuracy, and introducing a new metric that correlates well with the accuracy of the quantized model. Finally, we examine the impact of optimizer choice for quantization-aware training (QAT), and demonstrate that the optimizer performing best in higher precision may not remain optimal in the quantized setting. We further support our finding by deriving scaling laws for each optimizer under QAT.

There are still opportunities to further extend our study. We plan to investigate additional bitwidths (e.g., 8-bit and 6-bit) for QAT, as well as alternative PTQ schemes, and to study the error propagation of these methods using our ABC decomposition framework from Section 3.2. Studying the error propagation of other 4-bit data-types, like micro-scaling format, is also interesting. Moreover, deriving the gain decomposition analysis for other modules in the network (like self-attention) would be a very natural extension of our work. Lastly, given that many optimizers (e.g., PSGD, Muon) internally solve a constrained optimization problem, by adapting the problem to also include a condition about the gain $G_\ell$, so that the final weights propagate quantization error only mildly, one might be able to obtain a quantization-friendly optimizer.

### ACKNOWLEDGEMENTS

This work was supported by the ERC PSAP project (grant agreement No. 101002047), and under project ID "FlexLM: Efficient Targeted LLM Compression" as part of the Swiss AI Initiative, through a grant from the ETH Domain and computational resources provided by the Swiss National Supercomputing Centre (CSCS) under the Alps infrastructure.

This work was supported under project ID A140 as part of the Swiss AI Initiative, through a grant from the ETH Domain and computational resources provided by the Swiss National Supercomputing Centre (CSCS) under the Alps infrastructure.

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

# A APPENDIX

## A.1 ADDITIONAL FULL-PRECISION RESULTS

| Optimizer | Avg. Zero-Shot Acc.(↑) \|Test Loss(↓) | | | | | |
|---|---|---|---|---|---|---|
| | 50M | 125M | 350M | 500M | 760M | 1.5B |
| AdamW | 43.75 \| 3.695 | 48.64 \| 3.318 | 56.58 \| 2.961 | 60.39 \| 2.834 | 63.9 \| 2.744 | 67.93 \| 2.627 |
| Muon | 45.03 \| 3.632 | 49.62 \| 3.263 | **58.08** \| **2.915** | **61.86** \| **2.804** | **64.63** \| **2.719** | **69.19** \| **2.612** |
| PSGD | 45.08 \| 3.615 | 49.95 \| 3.283 | 55.99 \| 2.978 | 60.85 \| 2.841 | 61.47 \| 2.799 | N/A |
| Scion | 45.41 \| 3.615 | 49.83 \| 3.269 | 57.53 \| 2.932 | 61.68 \| 2.805 | 63.82 \| 2.726 | N/A |
| Shampoo | 44.81 \| 3.648 | 49.53 \| 3.311 | 56.51 \| 2.959 | 61.03 \| 2.831 | 63.05 \| 2.741 | 68.16 \| 2.622 |
| SOAP | **45.73** \| **3.589** | **50.24** \| **3.241** | 58.07 \| 2.916 | 61.34 \| 2.803 | 62.27 \| 2.754 | N/A |

Table 5: Results for the best models trained with BFloat16. For each model, we show the average zero-shot accuracy (↑) on the left, and the test loss (↓) on the right. As expected, the two track each other closely. While SOAP does better for small sizes, Muon is consistently the best for bigger models.

## A.2 ADDITIONAL POST-TRAINING QUANTIZATION RESULTS

| Optimizer | Model Size | | | | | |
|---|---|---|---|---|---|---|
| | 50M | 125M | 350M | 500M | 760M | 1.5B |
| AdamW | 5.445 | 4.233 | 4.274 | **3.730** | 3.480 | 3.769 |
| Muon | 4.305 | 4.145 | 4.941 | 4.819 | 4.928 | 9.725 |
| PSGD | 4.102 | **3.704** | 3.948 | 3.878 | 3.776 | N/A |
| Scion | 4.116 | 3.803 | 3.951 | 4.471 | 3.759 | N/A |
| Shampoo | 4.731 | 4.786 | **3.246** | 5.219 | **3.217** | **3.639** |
| SOAP | **4.027** | 3.790 | 4.222 | 3.987 | 6.320 | N/A |

Table 6: Test loss (↓) after we applied PTQ (row-wise W4A4) to the networks with the same common-loss (CL). The results mostly mirror Table 3, with Shampoo generally showing the mildest degradation. The exception is 500M, where Shampoo exhibits a noticeable degradation in test loss, despite outperforming other optimizers in downstream tasks.

| Optimizer | Model Size (GPTQ) |
|---|---|
| | 500M |
| AdamW | 55.47 |
| Muon | 50.22 |
| PSGD | 53.73 |
| Scion | 52.21 |
| Shampoo | **55.84** |
| SOAP | 51.16 |

Table 7: Average zero-shot accuracy (↑) after we applied GPTQ (W4A4) to the networks with the same common-loss (CL). The trends are similar to Table 3 (RTN).

| Optimizer | Model Size (Llama 2) |
|---|---|
| | 350M |
| AdamW | 47.35 |
| Muon | 45.20 |
| PSGD | 48.01 |
| Scion | 47.77 |
| Shampoo | **50.86** |
| SOAP | 48.13 |

Table 8: Average zero-shot accuracy (↑) after we applied PTQ (row-wise W4A4) to Llama 2 networks with common-loss 2.74. We see the same trends as with OLMo 2 (Table 3).

## A.3 ADDITIONAL QUANTIZATION-AWARE TRAINING RESULTS

| Optimizer | Model Size | | | | | |
|---|---|---|---|---|---|---|
| | 50M | 125M | 350M | 500M | 760M | 1.5B |
| AdamW | 3.757 | 3.375 | 3.009 | 2.905 | 2.787 | 2.655 |
| Muon | 3.698 | 3.340 | **2.971** | 2.868 | 2.765 | 2.651 |
| PSGD | 3.696 | 3.339 | 3.19 | 2.976 | 2.844 | N/A |
| Scion | 3.703 | 3.335 | 2.980 | 2.850 | **2.763** | N/A |
| Shampoo | 3.735 | 3.341 | 2.999 | 2.849 | 2.782 | **2.640** |
| SOAP | **3.682** | **3.302** | 2.981 | **2.842** | 2.797 | N/A |

Table 9: Final test loss (↓) for the W4A4 QAT models. Comparing with Table 5, we see that Shampoo shows the smallest degradation relative to the full-precision baselines.

| Optimizer | Model Size ($\frac{D}{N} = 5$) |
|---|---|
| | 350M |
| AdamW | 51.30 (-3.69) |
| Muon | 52.75 (-5.05) |
| PSGD | 49.68 (-7.88) |
| Scion | 53.23 (-2.71) |
| Shampoo | 52.51 (**-2.54**) |
| SOAP | **53.45** (-2.66) |

Table 10: Average zero-shot accuracy (↑) for the models trained with 4-bit QAT via QuEST, and data-to-model ratio of 5 instead of the Chinchilla-optimal 20. Shampoo shows the smallest quantization degradation, even though it does not lead to the best model at this scale (similarly to Table 4).

## A.4 ABC DECOMPOSITION

Consider a Feed-Forward Network (FFN) with $L$ modules $f_\ell(\cdot)$, for $\ell \in \{1, \ldots, L\}$. The output of the network $f(\boldsymbol{x}) \in \mathbb{R}^{n_L}$ for an input $\boldsymbol{x} \in \mathbb{R}^{n_0}$ is:

$$\boldsymbol{h}_0 = \boldsymbol{x}$$
$$\boldsymbol{h}_\ell = f_\ell(\boldsymbol{h}_{\ell-1}) \text{ for } \ell \in \{1, \ldots, L\}$$
$$f(\boldsymbol{x}) = \boldsymbol{h}_L$$

where $\boldsymbol{h}_\ell \in \mathbb{R}^{n_\ell}$ are the activations of the network.

The modules $f_\ell(\cdot)$ can be any arbitrary transformation, including linear layers, convolutions, activation functions, layer normalizations and self-attention. Thus, the class of networks we consider is quite general, and includes some of the most popular architectures, like Convolutional Networks (CNNs) and Transformers. Moreover, this framework is also flexible in terms of the granularity of the module partitioning. Even if a module also consists itself of submodules, we can arbitrarily choose to consider it as a whole. Therefore, we can study the network in any level of granularity desired.

Assume now that we "quantize" the modules, changing them from $f_\ell(\cdot)$ to $f_\ell^{\text{q}}(\cdot)$. There is again flexibility on what "quantize" can mean in this context. For example, we can arbitrarily choose to apply weight and activation quantization on linear layers, activation quantization on activation functions, and no quantization on layer normalizations. Other choices are also possible, as we treat "quantization" as a general perturbation in the function space.

After quantizing, the activations of the network become:

$$\boldsymbol{h}_0^{\text{q}} = \boldsymbol{x}$$
$$\boldsymbol{h}_\ell^{\text{q}} = f_\ell^{\text{q}}(\boldsymbol{h}_{\ell-1}^{\text{q}}) \text{ for } \ell \in \{1, \ldots, L\}$$
$$f^{\text{q}}(\boldsymbol{x}) = \boldsymbol{h}_L^{\text{q}}$$

By comparing the activations pre and post-quantization:

$$\boldsymbol{h}_\ell = f_\ell(\boldsymbol{h}_{\ell-1})$$
$$\boldsymbol{h}_\ell^{\text{q}} = f_\ell^{\text{q}}(\boldsymbol{h}_{\ell-1}^{\text{q}})$$

we see that the change in the output $\boldsymbol{h}_\ell$ is caused by the change in the input $\boldsymbol{h}_{\ell-1}$ (activation space), as well as the change in the function $f_\ell(\cdot)$ (function space). The change in the input $\boldsymbol{h}_{\ell-1}$ comes from the propagation of the quantization error through the previous $(\ell - 1)$ modules, while the change in the function $f_\ell(\cdot)$ is the newly introduced layerwise perturbation.

We wish to precisely quantify these changes. In order to do this, we study the difference in the activations:

$$\Delta \boldsymbol{h}_\ell := \boldsymbol{h}_\ell^{\text{q}} - \boldsymbol{h}_\ell = f_\ell^{\text{q}}(\boldsymbol{h}_{\ell-1}^{\text{q}}) - f_\ell(\boldsymbol{h}_{\ell-1})$$

By adding and subtracting either $f_\ell^{\text{q}}(\boldsymbol{h}_{\ell-1})$, or $f_\ell(\boldsymbol{h}_{\ell-1}^{\text{q}})$, we can get:

$$\Delta \boldsymbol{h}_\ell = (f_\ell^{\text{q}}(\boldsymbol{h}_{\ell-1}^{\text{q}}) - f_\ell^{\text{q}}(\boldsymbol{h}_{\ell-1})) + (f_\ell^{\text{q}}(\boldsymbol{h}_{\ell-1})) - f_\ell(\boldsymbol{h}_{\ell-1}))$$
$$\Delta \boldsymbol{h}_\ell = (f_\ell^{\text{q}}(\boldsymbol{h}_{\ell-1}^{\text{q}}) - f_\ell(\boldsymbol{h}_{\ell-1}^{\text{q}})) + (f_\ell(\boldsymbol{h}_{\ell-1}^{\text{q}}) - f_\ell(\boldsymbol{h}_{\ell-1}))$$

It is instructive to understand what the four terms on the right sides represent. The top left term stands for the change in the input under the function $f_\ell^{\text{q}}(\cdot)$, the top right term for the change in the function under the input $\boldsymbol{h}_{\ell-1}$, the bottom left for the change in the function under the input $\boldsymbol{h}_{\ell-1}^{\text{q}}$, and the last term for the change in the input under the function $f_\ell(\cdot)$.

Thus, the top equation implies that we first perturb $f_\ell(\cdot)$ to $f_\ell^{\text{q}}(\cdot)$, and then $\boldsymbol{h}_{\ell-1}$ to $\boldsymbol{h}_{\ell-1}^{\text{q}}$. On the other hand, the bottom equation implies that we first perturb the input, and then the function. Both equations are valid, but reflect different assumptions about the order of the perturbations. Arbitrarily choosing one can lead to attribution bias, and thus, following the Shapley principle[5], we average both

---

[5]According to the Shapley principle, the *unique* way to distribute contributions fairly across two players (here: input vs. function) is by averaging the possible orderings. This is the only allocation scheme that satisfies the three axioms of symmetry, efficiency and linearity.

as follows:

$$\boldsymbol{\Delta h}_\ell = \frac{(f_\ell^{\mathrm{q}}(\boldsymbol{h}_{\ell-1}^{\mathrm{q}}) - f_\ell^{\mathrm{q}}(\boldsymbol{h}_{\ell-1})) + (f_\ell^{\mathrm{q}}(\boldsymbol{h}_{\ell-1})) - f_\ell(\boldsymbol{h}_{\ell-1}))}{2} +$$

$$+ \frac{(f_\ell^{\mathrm{q}}(\boldsymbol{h}_{\ell-1}^{\mathrm{q}}) - f_\ell(\boldsymbol{h}_{\ell-1}^{\mathrm{q}})) + (f_\ell(\boldsymbol{h}_{\ell-1}^{\mathrm{q}}) - f_\ell(\boldsymbol{h}_{\ell-1}))}{2} =$$

$$= \frac{(f_\ell^{\mathrm{q}}(\boldsymbol{h}_{\ell-1}^{\mathrm{q}}) - f_\ell^{\mathrm{q}}(\boldsymbol{h}_{\ell-1})) + (f_\ell(\boldsymbol{h}_{\ell-1}^{\mathrm{q}}) - f_\ell(\boldsymbol{h}_{\ell-1}))}{2} +$$

$$+ \frac{(f_\ell^{\mathrm{q}}(\boldsymbol{h}_{\ell-1}^{\mathrm{q}}) - f_\ell(\boldsymbol{h}_{\ell-1}^{\mathrm{q}})) + (f_\ell^{\mathrm{q}}(\boldsymbol{h}_{\ell-1})) - f_\ell(\boldsymbol{h}_{\ell-1}))}{2} =$$

$$= \boldsymbol{a}_\ell + \boldsymbol{b}_\ell$$

with:

$$\boldsymbol{a}_\ell := \frac{(f_\ell^{\mathrm{q}}(\boldsymbol{h}_{\ell-1}^{\mathrm{q}}) - f_\ell^{\mathrm{q}}(\boldsymbol{h}_{\ell-1})) + (f_\ell(\boldsymbol{h}_{\ell-1}^{\mathrm{q}}) - f_\ell(\boldsymbol{h}_{\ell-1}))}{2}$$

$$\boldsymbol{b}_\ell := \frac{(f_\ell^{\mathrm{q}}(\boldsymbol{h}_{\ell-1}^{\mathrm{q}}) - f_\ell(\boldsymbol{h}_{\ell-1}^{\mathrm{q}})) + (f_\ell^{\mathrm{q}}(\boldsymbol{h}_{\ell-1})) - f_\ell(\boldsymbol{h}_{\ell-1}))}{2}$$

Hence, the change in the output $\boldsymbol{\Delta h}_\ell := \boldsymbol{h}_\ell^{\mathrm{q}} - \boldsymbol{h}_\ell$ can be *exactly* decomposed into two terms. The first term, $\boldsymbol{a}_\ell$ entirely captures the effect of the change in the input from $\boldsymbol{h}_{\ell-1}$ to $\boldsymbol{h}_{\ell-1}^{\mathrm{q}}$, while the second term $\boldsymbol{b}_\ell$ completely encodes the change in the function from $f_\ell(\cdot)$ to $f_\ell^{\mathrm{q}}(\cdot)$.

However, $\boldsymbol{\Delta h}_\ell, \boldsymbol{a}_\ell, \boldsymbol{b}_\ell$ are all still vectors. In order to reduce them to interpretable numbers, we compute the $L_2$-norm[6] of $\boldsymbol{\Delta h}_\ell$. Specifically, since $\|\boldsymbol{\Delta h}_\ell\|$ should be interpreted relative to the scale of the initial unquantized activations, we normalize with respect to $\|\boldsymbol{h}_\ell\|$. Thus, we study the $L_2$-norm of the relative change in the activations:

$$r_\ell := \frac{\|\boldsymbol{\Delta h}_\ell\|}{\|\boldsymbol{h}_\ell\|} = \frac{\|\boldsymbol{a}_\ell + \boldsymbol{b}_\ell\|}{\|\boldsymbol{h}_\ell\|} \iff r_\ell^2 = \frac{\|\boldsymbol{a}_\ell + \boldsymbol{b}_\ell\|^2}{\|\boldsymbol{h}_\ell\|^2}$$

By setting $R_\ell := r_\ell^2$ and using the Law of Cosines:

$$R_\ell = \frac{\|\boldsymbol{a}_\ell\|^2 + 2\langle\boldsymbol{a}_\ell, \boldsymbol{b}_\ell\rangle + \|\boldsymbol{b}_\ell\|^2}{\|\boldsymbol{h}_\ell\|^2} =$$

$$= \left(\frac{\|\boldsymbol{a}_\ell\|}{\|\boldsymbol{h}_\ell\|}\right)^2 + 2\frac{\langle\boldsymbol{a}_\ell, \boldsymbol{b}_\ell\rangle}{\|\boldsymbol{h}_\ell\|^2} + \left(\frac{\|\boldsymbol{b}_\ell\|}{\|\boldsymbol{h}_\ell\|}\right)^2 =$$

$$= A_\ell + B_\ell + C_\ell$$

where:

$$A_\ell := \left(\frac{\|\boldsymbol{a}_\ell\|}{\|\boldsymbol{h}_\ell\|}\right)^2$$

$$B_\ell := \left(\frac{\|\boldsymbol{b}_\ell\|}{\|\boldsymbol{h}_\ell\|}\right)^2$$

$$C_\ell := 2\frac{\langle\boldsymbol{a}_\ell, \boldsymbol{b}_\ell\rangle}{\|\boldsymbol{h}_\ell\|^2}$$

with $A_\ell$ standing for the quantization error accumulated from the previous $(\ell-1)$ modules, $B_\ell$ for the new quantization error, introduced in the $\ell$-th layer, and $C_\ell$ for the interaction between the two.

A quantity of interest that helps us understand how a module amplifies or attenuates the quantization error from the previous layer is the gain:

$$G_\ell := \frac{A_\ell}{R_{\ell-1}}$$

---

[6]We can also use the "more natural" RMS-norm (as in Large et al. (2024)), which is normalized with respect to dimension, allowing us to compare numbers across networks with different widths. In the end, these two choices are equivalent, because we ultimately investigate ratios of $L_2$-norms, which are equal to ratios of RMS-norms.

$G_\ell$ determines how the quantization error of the previous layer, expressed by $R_{\ell-1}$, propagates through the $\ell$-th module to appear as $A_\ell$.

In general, if we treat the module as a black-box, we cannot further analyze the gain $G_\ell$ (even though we can calculate it from $A_\ell$ and $R_{\ell-1}$). However, for specific functional forms of $f_\ell(\cdot)$ we can exactly recover $G_\ell$ in closed-form. The most straightforward example is the case of a linear layer under weight and activation quantization.

Specifically, for a linear layer[7] under joint quantization:

$$\boldsymbol{h}_\ell = f_\ell(\boldsymbol{h}_{\ell-1}) = \boldsymbol{W}_\ell \boldsymbol{h}_{\ell-1}$$
$$\boldsymbol{h}_\ell^{\mathrm{q}} = f_\ell^{\mathrm{q}}(\boldsymbol{h}_{\ell-1}^{\mathrm{q}}) = (\boldsymbol{W}_\ell + \varepsilon_\ell^W)\boldsymbol{h}_{\ell-1}^{\mathrm{q}} + \varepsilon_\ell^h$$

where $\varepsilon_\ell^W$ and $\varepsilon_\ell^h$ are quantization noise introduced by weight and activation quantization[8], respectively.

Moreover:

$$f_\ell^{\mathrm{q}}(\boldsymbol{h}_{\ell-1}) = (\boldsymbol{W}_\ell + \varepsilon_\ell^W)\boldsymbol{h}_{\ell-1} + \varepsilon_\ell^h$$
$$f_\ell(\boldsymbol{h}_{\ell-1}^{\mathrm{q}}) = \boldsymbol{W}_\ell \boldsymbol{h}_{\ell-1}^{\mathrm{q}}$$

By using the last four equations:

$$f_\ell^{\mathrm{q}}(\boldsymbol{h}_{\ell-1}^{\mathrm{q}}) - f_\ell^{\mathrm{q}}(\boldsymbol{h}_{\ell-1}) = (\boldsymbol{W}_\ell + \varepsilon_\ell^W)\boldsymbol{\Delta}\boldsymbol{h}_{\ell-1}$$
$$f_\ell(\boldsymbol{h}_{\ell-1}^{\mathrm{q}}) - f_\ell(\boldsymbol{h}_{\ell-1}) = \boldsymbol{W}_\ell \boldsymbol{\Delta}\boldsymbol{h}_{\ell-1}$$
$$f_\ell^{\mathrm{q}}(\boldsymbol{h}_{\ell-1}^{\mathrm{q}}) - f_\ell(\boldsymbol{h}_{\ell-1}^{\mathrm{q}}) = \varepsilon_\ell^W \boldsymbol{h}_{\ell-1}^{\mathrm{q}} + \varepsilon_\ell^h$$
$$f_\ell^{\mathrm{q}}(\boldsymbol{h}_{\ell-1})) - f_\ell(\boldsymbol{h}_{\ell-1}) = \varepsilon_\ell^W \boldsymbol{h}_{\ell-1} + \varepsilon_\ell^h$$

In turn, by replacing those into the definitions of $\boldsymbol{a}_\ell$ and $\boldsymbol{b}_\ell$ we obtain:

$$\boldsymbol{a}_\ell = (\boldsymbol{W}_\ell + \frac{1}{2}\varepsilon_\ell^W)\boldsymbol{\Delta}\boldsymbol{h}_{\ell-1}$$
$$\boldsymbol{b}_\ell = \varepsilon_\ell^W \left(\frac{\boldsymbol{h}_{\ell-1} + \boldsymbol{h}_{\ell-1}^{\mathrm{q}}}{2}\right) + \varepsilon_\ell^h$$

Finally, by using the definition of $A_\ell$:

$$A_\ell := \left(\frac{\|\boldsymbol{a}_\ell\|}{\|\boldsymbol{h}_\ell\|}\right)^2 =$$
$$= \left(\frac{\|(\boldsymbol{W}_\ell + \frac{1}{2}\varepsilon_\ell^W)\boldsymbol{\Delta}\boldsymbol{h}_{\ell-1}\|}{\|\boldsymbol{W}_\ell \boldsymbol{h}_{\ell-1}\|}\right)^2 =$$
$$= \left(\frac{\|\boldsymbol{W}_\ell + \frac{1}{2}\varepsilon_\ell^W\|_* \|\boldsymbol{\Delta}\boldsymbol{h}_{\ell-1}\| \cos\phi_\ell}{\|\boldsymbol{W}_\ell\|_* \|\boldsymbol{h}_{\ell-1}\| \cos\psi_\ell}\right)^2 = \quad (\|\cdot\|_* \text{ denotes the spectral norm})$$
$$= \left(\frac{\|\boldsymbol{W}_\ell + \frac{1}{2}\varepsilon_\ell^W\|_*}{\|\boldsymbol{W}_\ell\|_*}\right)^2 \left(\frac{\|\boldsymbol{\Delta}\boldsymbol{h}_{\ell-1}\|}{\|\boldsymbol{h}_{\ell-1}\|}\right)^2 \left(\frac{\cos\phi_\ell}{\cos\psi_\ell}\right)^2 =$$
$$= G_{1,\ell} G_{2,\ell} R_{\ell-1} \iff$$
$$\iff G_\ell = G_{1,\ell} G_{2,\ell}$$

---

[7]We assume no bias for ease of exposition (this also coincides with our architecture).

[8]Here, without loss of generality, we assume we quantize the activations after the matrix multiplication. Quantizing before the matrix multiplication would mean $\boldsymbol{h}_\ell^{\mathrm{q}} = (\boldsymbol{W}_\ell + \varepsilon_\ell^W)(\boldsymbol{h}_{\ell-1}^{\mathrm{q}} + \varepsilon_\ell^h) = (\boldsymbol{W}_\ell + \varepsilon_\ell^W)\boldsymbol{h}_{\ell-1}^{\mathrm{q}} + (\boldsymbol{W}_\ell + \varepsilon_\ell^W)\varepsilon_\ell^h$, which would lead to equivalent analysis, since we can treat the last term as a single quantity.

where we defined:

$$\cos \phi_\ell := \frac{\|(\boldsymbol{W}_\ell + \frac{1}{2}\boldsymbol{\varepsilon}_\ell^W)\boldsymbol{\Delta h}_{\ell-1}\|}{\|\boldsymbol{W}_\ell + \frac{1}{2}\boldsymbol{\varepsilon}_\ell^W\|_*\|\boldsymbol{\Delta h}_{\ell-1}\|}$$

$$\cos \psi_\ell := \frac{\|\boldsymbol{W}_\ell \boldsymbol{h}_{\ell-1}\|}{\|\boldsymbol{W}_\ell\|_*\|\boldsymbol{h}_{\ell-1}\|}$$

$$G_{1,\ell} := \left(\frac{\|\boldsymbol{W}_\ell + \frac{1}{2}\boldsymbol{\varepsilon}_\ell^W\|_*}{\|\boldsymbol{W}_\ell\|_*}\right)^2 \quad (\text{``Spectral ratio''})$$

$$G_{2,\ell} := \left(\frac{\cos \phi_\ell}{\cos \psi_\ell}\right)^2 \quad (\text{``Alignment ratio''})$$

The angles $\phi_\ell$ and $\psi_\ell$ are between matrices and vectors. This is a non-standard definition, but quite intuitive nonetheless. The angles lie between $0°$ ($\cos \phi_\ell/\psi_\ell = 1$) and $90°$ ($\cos \phi_\ell/\psi_\ell = 0$), and indicate how close is the direction of the vector to the direction of the top right singular vector of the matrix. An angle of $0°$ means that the vector points towards a direction that is maximally stretched by the matrix. In other words, the vector is colinear with the top right singular vector. In contrast, an angle of $90°$ indicates that the vector and top right singular vector are orthogonal, and that the vector is being shrunk to 0.

The angle $\phi_\ell$ is between the change in the activations $\boldsymbol{\Delta h}_{\ell-1}$ and the weight matrix after quantization ($\boldsymbol{W}_\ell + \frac{1}{2}\boldsymbol{\varepsilon}_\ell^W$) (the $\frac{1}{2}$ comes from Shapley). The angle $\psi_\ell$ is between the incoming activations $\boldsymbol{h}_{\ell-1}$ and the unquantized weight matrix $\boldsymbol{W}_\ell$, and denotes how aligned are the incoming activations with the weights. The "spectral ratio" $G_{1,\ell}$ is the (squared) ratio of the spectral norms of weights before and after quantization. Finally, the "alignment ratio" $G_{2,\ell}$ just encodes the (squared) ratio of the alignment between the change in the activations and the weights after quantization, and the alignment between the incoming activations and the original weights.

To sum up, for a linear layer under joint quantization, we further factored the gain $G_\ell$ of quantization error into a product of two interpretable terms, the spectral ratio $G_{1,\ell}$, and the "alignment ratio" $G_{2,\ell}$. We can do a similar analysis for the terms $B_\ell$ and $C_\ell$, which we omit here because we found that $R_\ell$ is dominated by $A_\ell$ (see Figure 3). A similar-style analysis can also be done for other modules, like layer normalizations and self-attention, by using Taylor's theorem, but we leave this for future work.

## A.5 Optimizers' memory and computational complexities

Table 11 shows the asymptotic memory (including gradients) and computational complexities for a one step of gradient update of a hidden layer with weight matrix $W \in \mathbb{R}^{m \times n}$.

| Optimizer | Memory Overhead | Computational Overhead |
|---|---|---|
| **AdamW** | $3mn$ | $O(mn)$ |
| **Muon** | $2mn$ | $O(T(2nm^2 + m^3)), \ m \le n$ |
| **PSGD** | $mn + m^2 + n^2$ | $O(\frac{m^3+n^3}{f} + 2(m+n)mn)$ |
| **Scion** | $2mn$ | $O(T(2nm^2 + m^3)), \ m \le n$ |
| **Shampoo** | $3mn + m^2 + n^2$ | $O((m^3 + n^3)(1 + \frac{1}{f}) + (m+n)mn)$ |
| **SOAP** | $3mn + m^2 + n^2$ | $O((m^3 + n^3)(1 + \frac{1}{f}) + 2(m+n)mn)$ |

Table 11: Asymptotic memory (including gradients) and computational complexities **per one optimizer step** for a linear layer of size $m \times n$. Here $T$ denotes the number of Newton-Schulz iterations in Muon algorithm, and $f$ is the preconditioner update frequency for PSGD, Shampoo, and SOAP.

Recall that multiplication of 2 matrices $AB$, where $A \in \mathbb{R}^{m \times k}$, $B \in \mathbb{R}^{k \times n}$, requires $mnk$ multiplications and $mn(k - 1)$ additions, which totals in $mn(k + k - 1) \approx 2mnk$ operations.

| Optimizer | Iteration time (s) \| Memory (GB) | | | | | |
|---|---|---|---|---|---|---|
| | 50M (128) | 125M (64) | 350M (64) | 500M (32) | 760M (32) | 1.5B (16) |
| AdamW | 0.66 \| 63 | 1.37 \| 41 | 3.07 \| 66 | 4.51 \| 48 | 6.28 \| 57 | 12.89 \| 66 |
| Muon | 0.70 \| 63 | 1.41 \| 41 | 3.11 \| 66 | 4.59 \| 48 | 6.35 \| 57 | 13.09 \| 61 |
| PSGD | 0.80 \| 63 | 1.70 \| 41 | 3.49 \| 66 | 5.12 \| 50 | 7.08 \| 63 | N/A |
| Scion | 0.71 \| 63 | 1.51 \| 41 | 3.14 \| 66 | 4.63 \| 48 | 6.55 \| 57 | N/A |
| Shampoo | 1.30 \| 67 | 2.18 \| 48 | 4.34 \| 83 | 6.43 \| 72 | 8.38 \| 89 | 16.38 \| 91 |
| SOAP | 0.82 \| 63 | 1.52 \| 42 | 3.34 \| 66 | 5.02 \| 58 | 7.02 \| 77 | N/A |

Table 12: Measurements for full-precision training on a GH200 96GB GPU. Inside the parentheses next to each model size, we list the maximum batch size that works with all optimizers, also used for the measurements. AdamW and Muon have similar requirements, despite Muon leading to the strongest models (see Table 2). Even though models trained with Shampoo show the mildest degradation (see Table 3), they also take the longest to train, while also requiring the most memory.

1. **AdamW**
   Gradients + 1 and 2 moments = $3mn$ memory. The number of moments and weight update operations is linear with respect to $mn$, the number of elements in the weight matrix.

2. **Muon**
   Gradients ($mn$) and the first moment ($mn$) give the total memory overhead of $2mn$.

   Suppose $m < n$, one step of the Newton-Schulz iteration for matrix $X \in \mathbb{R}^{m \times n}$ then becomes

   $$X' = aX + b(XX^T)X + c(XX^T)^2 X, \tag{3}$$

   which gives $O(mn + 2m^2n + 2m^3 + 2m^2n) = O(2m^2n + m^3)$ time complexity. For the total of $T$ iterations it then becomes $O(T(2m^2n + m^3))$.

3. **PSGD**
   Gradients ($mn$) + 2 preconditioning matrices ($m \times m$ and $n \times n$) give $mn + m^2 + n^2$ total memory.

   Applying preconditioners $P_L \in \mathbb{R}^{m \times m}$ and $P_R \in \mathbb{R}^{n \times n}$ on both sides of the gradient $P_L g P_R$ uses $2m^2n + 2mn^2$ operations, and the preconditioner update has computational complexity $O(m^3 + n^3)$. Amortized by update frequency $f$, the total complexity is $O(2m^2n + 2mn^2 + \frac{m^3+n^3}{f})$.

4. **Scion**

   Gradients ($mn$) and the first moment ($mn$) give the total memory overhead of $2mn$.

   Scion uses the same Spectral LMO as Muon, which results in $O(T(2m^2n + m^3))$ complexity overhead.

5. **Shampoo**

   Memory overhead consists of gradients, 1st and 2nd moments ($3mn$), left and right preconditioners and eigenvector matrices ($2m^2 + 2n^2$).

   Updating the preconditioners takes $O(m^3 + n^3)$, and applying them additional $O(m^2n + mn^2)$ operations. Preconditioner change with frequency $f$ adds extra $O(\frac{m^3+n^3}{f})$ complexity. In practice $f \sim 10 - 100$, so other terms dominate.

6. **SOAP**

   The same as for Shampoo, the total memory complexity is $3mn + 2m^2 + 2n^2$.

   The time complexity is same as for Shampoo with extra $O(m^2n + mn^2)$ for projecting back from the eigen-space. Recomputing the eigenbasis $Q_L, Q_R$ with frequency $f$ adds extra $O(\frac{m^3+n^3}{f})$ complexity. In practice $f \sim 10 - 100$, so other terms dominate.

A.6   SCALING LAWS FITTING PROCEDURE

To estimate the parameters of scaling law for each optimizer, we perform nonlinear least-squares regression using `scipy.least_squares`. To improve robustness against outliers, we employ the Huber loss with parameter $\delta = 10^{-3}$. We minimize the residuals between the predicted and observed log-loss values, fitting $f(\log N_i, \log D_i, \rho_{b_i})$ to $\log L_i$, where $N_i$ is the model size, $D_i$ the dataset size, and $b_i$ is the precision. The loss values are transformed into logarithmic scale prior to fitting for numerical stability. For the total parameter count $N$, we include embedding parameters, which we found improves the fit. We report the values with confidence intervals, obtained by leave-one-out cross-validation, in Table 13.

We estimated the confidence intervals of the fitted hyperparameters by computing their standard deviations across folds in a leave-one-out cross-validation procedure. We found that excluding the loss value corresponding to the largest model size provides an unstable fit, so we retained the lowest loss point for all the folds.

| Optimizer | $A'$ | $\alpha$ | $E$ | $\rho_{\text{4bit}}$ |
|---|---|---|---|---|
| AdamW | $79_{\pm 7}$ | $0.20_{\pm 0.01}$ | $1.40_{\pm 0.05}$ | $0.863_{\pm 0.003}$ |
| Muon | $208_{\pm 40}$ | $0.27_{\pm 0.01}$ | $1.85_{\pm 0.04}$ | $0.852_{\pm 0.010}$ |
| PSGD | $77_{\pm 6}$ | $0.18_{\pm 0.05}$ | $1.39_{\pm 0.44}$ | $0.739_{\pm 0.049}$ |
| Scion | $148_{\pm 22}$ | $0.25_{\pm 0.01}$ | $1.75_{\pm 0.07}$ | $0.856_{\pm 0.010}$ |
| Shampoo | $142_{\pm 26}$ | $0.24_{\pm 0.01}$ | $1.72_{\pm 0.09}$ | $0.879_{\pm 0.018}$ |
| SOAP | $706_{\pm 132}$ | $0.35_{\pm 0.01}$ | $2.22_{\pm 0.03}$ | $0.822_{\pm 0.010}$ |

Table 13: Scaling law coefficients for different optimizers.

