# OpenReview forum: "Beyond Outliers: A Study of Optimizers Under Quantization"
_ICLR.cc/2026/Conference — ICLR 2026 Poster_

### Official Review · Reviewer_Ag8F · 2025-10-29

**Soundness:** 3
**Presentation:** 4
**Contribution:** 3
**Rating:** 8
**Confidence:** 3

**Summary:**

The authors study the interaction between quantization and choice of optimizers for both PTQ and QAT scenarios. The authors train BF16 models with various optimizers and perform PTQ to measure the degration. The authors then propose a new metric which correlates well with the PTQ performance compared to commonly used metrics like MMD and Kurtosis. The authors then study the impact of optimizer choice during W4A4 QAT and show that the choice of optimizer does not transfer directly from high precision to low precision.

**Strengths:**

- This is a great paper. Extremely well written and addresses important questions still unresolved in the community.
- Compares models using downstream performance rather than a proxy metric like validation loss.
- The differences in error propagation through the network with various optimizers was an interesting observation.

**Weaknesses:**

- Only RTN PTQ scheme is studied, which is not usually used in practice.
- For QAT, microscaling formats like MXFP4/ NVFP4 would perhaps be a better choice
- Due to lack of using hyper parameter selection techniques like MuP, the current hyperparameter choices might be suboptimal.

**Questions:**

- Why is the OLMo2 architecture changed to incorporate $\text{ReLU}^2$?
- Any reason for not using MuP for hyperparameter selection?
- Is it reasonable to expect the optimal hyperparameters to be same for both the full-precision and low-precision runs?
- What happens if we use fancier PTQ schemes compared to Round-to-nearest? Is the result still the same?
- "Shampoo’s networks are also characterized by the highest MMR". Figure 2 suggests it's PSGD?

---

> ### Author Response · Authors · 2025-11-23
> **Reply to Review**
>
> Thanks a lot for your comments!
>
> Replies to Weaknesses:
> 1) **Only RTN**: To address this, we have extended our study to also include GPTQ for the 500M models. The results, shown in the table below, closely follow the RTN findings.
> |Optimizer|Avg. zero-shot, GPTQ (W4A4)|
> |---|---|
> |AdamW|55.47|
> |Muon|50.22|
> |PSGD|53.73|
> |Scion|52.21|
> |Shampoo|**55.84**|
> |SOAP|51.16|
> 2) **MXFP4 QAT**: While extending our study to other QAT methods would indeed be reassuring of the generality of our results, it is unfortunately beyond our computational budget. However, given that all other results (PTQ, theoretical, QuEST) show the same trends, we expect Shampoo's advantage to still hold for other QAT schemes. We also note that QuEST is currently the state-of-the-art method, consistently outperforming STE and LSQ.
> 3) **muP**: We do not and cannot use muP in its known formulations because: i) The muP learning rate coefficients have only been derived for Adam and Scion, and not the other four optimizers we are testing, ii) Crucially, muP only allows width transfer. In our setup, we increase the model size by increasing both width and depth (as commonly done in practice). Moreover, independent sweeping would cause the computational costs to explode combinatorially, making a study like this impossible. Besides, in most cases, the differences are wide enough, and consistent, that we think a more exhaustive hyperparameter search would not meaningfully alter our results.
>
> Questions:
> 1) We were initially planning to also investigate sparsity, and $\text{ReLU}^2$ has been known (https://arxiv.org/abs/2402.03804) to help there. However, when we realized that MMR/Kurtosis were not indicative of quantization performance, we shifted towards exploring this theoretically.
> 2) See (3) above.
> 3) In general, there is no reason to believe this to be the case apriori. Unfortunately, investigating this conclusively would vastly increase the computational cost of our study. Nevertheless, in limited and smaller scale experiments, this was indeed the case. Also, the fact that there is consistency between our results is reassuring.
> 4) See (1) above.
> 5) Figure 2 shows the full learning rate sweep. However, all of Section 3.2, including the sentence you cited, only involves the optimal learning rate (with respect to the validation loss). In other words, for Section 3.2, we only look at the orange point that corresponds to the minimum blue point in Figure 2. Among all these orange points, the highest one corresponds to Shampoo. So Shampoo indeed has the highest MMR.

---

> ### Author Response · Authors · 2025-11-27
> **Reply Update**
>
> Dear Reviewer,
>
> We updated the manuscript with the GPTQ results (shown above). We also improved the paper with the suggestions of the other reviewers (a summary of all the changes can be found as an Official Comment).
>
> As the discussion is soon drawing to a close, we wanted to send you a gentle reminder regarding the discussion. We would be very happy if you could please examine our responses and determine whether they addressed your concerns.
>
> Best regards,
>
> The authors

---

> > ### Comment · Reviewer_Ag8F · 2025-11-27
> >
> > Thank you for your response. I have already advocated for acceptance; I maintain my score and stance.

---

### Official Review · Reviewer_7Mqb · 2025-11-01

**Soundness:** 3
**Presentation:** 3
**Contribution:** 2
**Rating:** 4
**Confidence:** 4

**Summary:**

This work explores the impact of quantization on various optimizers. This work presents a comprehensive empirical analysis of how optimizers fare before and after quantization. The authors dispute that using outlier-related metrics like max-to-mean ratio (MMR) of input tensors is a useful indicator of quantization performance. Instead, this work focuses on studying the relative change in activations as a more expressive quantity to observe. The experiments are performed on recent class of transformer-based models, using a wide array of recent/popular optimizers. This work concludes that based on their experiments Shampoo optimizer fares best with the lowest accuracy degradation.

**Strengths:**

* The stated goals of the work, of assessing the impact of quantization on different optimizers, and formalizing the accuracy degradation using better metrics than MMR is interesting.
* The work makes a compelling case showing that the focus on outliers is not useful to predict accuracy degradation. The formalism of relative change in activations $R_\ell$ as a proxy for total quantization error is convincing.
* Experiments are comprehensive; provide solid empirical evidence for the claims in the paper.

**Weaknesses:**

* **Outlier-centric motivation:** One of the main motivation of this empirical study is to show that outlier-based metrics are not predictive of accuracy degradation due to quantization. However, the authors do not point to any relevant literature or the general consensus in the community that argues this is the case. In my opinion, the consensus is on straight-through-estimator (STE) when using quantization aware training (QAT) [1], and uniform quantization that are the more widely cited reasons [2]. Can the authors point to the specific literature that claims outliers as the problem?

* **Motivation for pursuing gain**: The point above also leads to my next point. While empirical results show that relative change of activation, formulated as spectral ratio and alignment ratio, are more indicative of performance degradation, what is the theoretical justification for this? For instance, QAT has mechanisms to overcome these activation rescaling as it is performed during training, and has the opportunity to compensate for these fluctuations. Are there other subtle mechanisms that this work is identifying?

* **Oscillations in Figures 3 & 4:** The oscillations in Figure 3 and more pronounced in Figure 4 warrant some attention. What is going on in Fig. 4? Why are $G_\ell, R_\ell$ oscillating between regular values between successive layers? This seems to be the case for all optimizers.
* **Point of Scaling laws** While the analysis and the fit for the scaling laws are interesting, I was confused as to how this ties to the rest of the paper. Is it to show that $\rho$ for Shampoo is slightly higher than AdamW? What else do we learn from these scaling curves?

* **Use of OLMo2 architecture:** The experiments are all performed on the same architecture family. The experiments are across different number of parameters, but the results and observations in this paper might be overfitting to a single architecture family. What is the justification? And would the authors expect these observations to generalize to other architecture families?

* **Optimizer hyperparameters:** It is mentioned that all hyperparameters for the optimizer were chosen based on the 50M model. Were these done in any consistent manner? For instance, following the muP recommendations?[3] If not, what is the argument for doing this?

### Other comments

* Two versions of the same paper, Panferov et al. 2025, are referenced
* Difficult to see what is going on with the components of $R_\ell$ i.e, the ABC decomposed terms.
* MMR abbreviation is introduced several times.
* Column 2 in Table 1 is redundant, given the first column.

### References

[1] Huh, Minyoung, et al. "Straightening out the straight-through estimator: Overcoming optimization challenges in vector quantized networks." International Conference on Machine Learning. PMLR, 2023.

[2] Oh, Sangyun, et al. "Non-uniform step size quantization for accurate post-training quantization." European Conference on Computer Vision. Cham: Springer Nature Switzerland, 2022.

[3] Yang, Greg, et al. "Tensor programs v: Tuning large neural networks via zero-shot hyperparameter transfer." arXiv preprint arXiv:2203.03466 (2022).

**Questions:**

See points under weaknesses.

---

> ### Author Response · Authors · 2025-11-23
> **Reply to Review**
>
> We sincerely appreciate you taking the time to review our paper.
>
> Replies to Weaknesses:
> 1) **Outlier-centric motivation**: This was covered in part in the second paragraph of our submission’s introduction. There, among others, we cite papers [2-5], which state that outliers are indeed the main cause of quantization difficulties. (For example, Nrusimha et al. [4] presents a technique that works by kurtosis regularization.) The same works also study outliers' prevalence indirectly, via "tailedness" statistics, such as Kurtosis and MMR. However, we noticed that we indeed forgot to cite the very first work about outliers and quantization degradation, namely the very influential work of [1]. We will add this to the updated text.
> 2) **Motivation for pursuing gain**: In brief, the main motivation of the ABC decomposition is as follows: We want to study how far the activations of the quantized network $h^q$ are from the activations of the full-precision network $h$. We can arbitrarily choose any measure of distance, but arguably the most natural one is the Euclidean norm: $\|h^q-h\|$. The L2 norm allows a closed-form mathematical analysis, and linking with interpretable quantities (like the spectral/alignment ratios). Even though this quantity is already serviceable, we usually also want to be able to compare values across different layers (so that we study error propagation through the network). This is why we normalize with respect to the L2-norm of the original activations. Hence, we end up with the norm of the relative change of the activations due to quantization, which is also a natural measure of perturbation. This quantity also correlates much better with quantization degradation than MMR/Kurtosis, as we see in Figure 1. Lastly, most QAT methods only consider and account for the per-layer introduced errors (through rescaling). They thus ignore error propagation.
> In our paper, we identify that every layer does not only add error due to the quantization of the specific layer, but also amplifies the error coming from the previous layer. To our knowledge, this is a new insight, and could be extended for the design of QAT/PTQ methods.
> 3) **Oscillations in Figures 3 & 4**: As stated in Section 3.2 (l357-359), we treat layer normalizations, residual connections, linear layers, activation functions and MHSA modules separately. The regular dips in the plots correspond to RMSNorm layers, which attenuate the quantization error instead of amplifying it. Indeed, even though we would expect the quantization error to grow on average as we go through the network, there is no reason to expect this at every single module. Moreover, the fact that the dips in Figures 3 and 4 "agree" is reassuring, since they are computed separately, and their agreement indicates that the computation is indeed correct. We will update the captions to make this more clear.
> 4) **Point of Scaling laws**: The role of the scaling laws section is to show that every optimizer under QAT follows a power-law regime, which can be captured by introducing only one extra hyperparameter in the scaling law. This lets us express the impact of quantization by a single scalar $\rho$, from which we find that Shampoo suffers from the least quantization penalty among all optimizers we study, making it a promising choice for low-precision training. Overall, the fitted laws provide practical guidance: they allow to estimate the performance of different optimizer/model size configurations and thus inform the choice of configuration under a given compute budget.
> 5) **Choice of OLMo2**: The effect of different optimizers on the linear layers should be the same (see Figure 4), and thus we would expect similar results for alternative architectures. To confirm this, we are currently rerunning our 350M experiments for Llama 2 on FineWeb. We will update this thread, as well as the main text, as soon as we obtain our results (probably in a few days).
> 6) **Hyperparameter optimization**: As outlined in Section 2 (l129-139), we sweep the learning rate for **every** model size. We only fix the rest of the hyperparameters to the values discovered with the sequential 50m sweeps. We do not and cannot use muP in its known formulations because: i) The muP learning rate coefficients have only been derived for Adam and Scion, and not the other four optimizers we are testing, ii) Crucially, muP only allows width transfer. In our setup, we increase the model size by increasing both width and depth (as commonly done in practice).
>
> Other Comments:
>
> 1+2+3) We will fix it. Thanks for your feedback!
>
> 4\) Including the **exact** count of parameters can be very useful for reproducibility (e.g., confirming proper loading of the model weights).

---

> > ### Author Response · Authors · 2025-11-23
> > **Reply to Review - References**
> >
> > References in the first reply:
> >
> > [1] Dettmers, Tim, et al. "Gpt3. int8 (): 8-bit matrix multiplication for transformers at scale." Advances in neural information processing systems 35 (2022): 30318-30332.
> >
> > [2] Wei, Xiuying, et al. "Outlier suppression: Pushing the limit of low-bit transformer language models." Advances in Neural Information Processing Systems 35 (2022): 17402-17414.
> >
> > [3] Bondarenko, Yelysei, Markus Nagel, and Tijmen Blankevoort. "Quantizable transformers: Removing outliers by helping attention heads do nothing." Advances in Neural Information Processing Systems 36 (2023): 75067-75096.
> >
> > [4] Nrusimha, Aniruddha, et al. "Mitigating the impact of outlier channels for language model quantization with activation regularization." arXiv preprint arXiv:2404.03605 (2024).
> >
> > [5] He, Bobby, et al. "Understanding and minimising outlier features in neural network training." arXiv preprint arXiv:2405.19279 (2024).

---

> > > ### Author Response · Authors · 2025-11-27
> > > **Reply Update**
> > >
> > > Dear Reviewer,
> > >
> > > We updated the manuscript with:
> > > 1) The citation we mentioned above in (1)
> > > 2) Comments on the captions of Figures 3 and 4 about the oscillations.
> > > 3) Experiments on 350M Llama 2 models and FineWeb. The results are similar to the OLMo2 models:
> > > |Optimizer|Avg. zero-shot, Llama 2 (W4A4)|
> > > |---|---|
> > > |AdamW|47.35|
> > > |Muon|45.20|
> > > |PSGD|48.01|
> > > |Scion|47.77|
> > > |Shampoo|**50.86**|
> > > |SOAP|48.13|
> > > 4) Fixes for your "Other comments" (one version of Panferov et al. 2025, better caption for ABC decomposition of Figure 3, removed multiple definitions of MMR)
> > >
> > > We also improved the paper with the suggestions of the other reviewers (a summary of all the changes can be found as an Official Comment).
> > >
> > > As the discussion is soon drawing to a close, we wanted to send you a gentle reminder regarding the discussion. We would be very happy if you could please examine our responses and determine whether they addressed your concerns.
> > >
> > > Best regards,
> > >
> > > The authors

---

### Official Review · Reviewer_f8Mk · 2025-11-04

**Soundness:** 3
**Presentation:** 3
**Contribution:** 2
**Rating:** 4
**Confidence:** 3

**Summary:**

This paper addresses a critical gap in the intersection of large language model (LLM) optimization and quantization: how optimizer choice impacts model performance when quantization (a key technique for efficient LLM deployment) is applied. It systematically investigates two major quantization paradigms—Post-Training Quantization (PTQ) (quantizing a fully trained model) and Quantization-Aware Training (QAT) (integrating quantization during training)—across LLMs of varying sizes (50M to 1.5B parameters) trained with six optimizers (AdamW, Muon, PSGD, Scion, Shampoo, SOAP).

The study first establishes well-tuned full-precision (FP) baselines using the OLMo2 architecture (modified with tied input-output weights and ReLU² activation) and the Chinchilla-optimal data-to-model ratio. It then evaluates how these FP models degrade under PTQ, and how optimizers perform when training quantized models from scratch via QAT. A key theoretical contribution is the ABC decomposition framework to analyze quantization error propagation, which reveals limitations of traditional outlier-centric metrics (e.g., Max-to-Median Ratio (MMR), Kurtosis) in predicting PTQ performance.

**Strengths:**

1. As the first work to rigorously explore optimizer-quantization interactions, it covers a broad scope—models from 50M to 1.5B parameters, 6 optimizers (AdamW, Muon, etc.), and both PTQ/QAT paradigms. This breadth avoids narrow conclusions and ensures results are tested across diverse scenarios.

2. It establishes robust full-precision baselines using the Chinchilla-optimal data-to-model ratio, detailed hyperparameter tuning (e.g., sequential sweeps on 50M models), and clear architectural modifications (OLMo2 with tied weights/ReLU²). Evaluation uses standard zero-shot benchmarks (PIQA, HellaSwag) and open tools (LM Evaluation Harness), ensuring reproducibility.

**Weaknesses:**

1. Your paper focuses on validating optimizer-quantization interactions using the OLMo2 architecture (with modifications limited to tied input-output weights and the ReLU² activation function), which provides a clear and controlled experimental base. We’re curious, have you considered extending these experiments to more mainstream LLM architectures like Llama or GPT-2 (or even Llama 3, which is widely used in industry)? We’d also be interested to hear the reasoning behind choosing OLMo2, and how you anticipate Shampoo’s quantization advantages might hold (or adapt) when tested on these other architectures, to further strengthen the generalizability of your conclusions.

2. The proposed \(R_\ell\) metric effectively addresses the limitations of traditional metrics like MMR and Kurtosis by capturing layer-wise activation error propagation, which is a valuable theoretical contribution. We noted that computing \(R_\ell\) for large models (e.g., 1.5B parameters) requires iterating through all layers’ activation tensors, leading to higher computational complexity than MMR/Kurtosis. In real-world deployment, users often need fast quantization outcome predictions (e.g., during preprocessing). We’re wondering if you have considered ways to optimize \(R_\ell\)’s computational cost, such as approximating it with key layers rather than all layers, and how you balance the metric’s predictive accuracy with the industry’s demand for low-cost tools?

3. Your QAT experiments use the QuEST scheme, which is a strong choice for stable low-precision training. Given that industry frequently adopts other QAT variants like GPTQ and AWQ (which differ in handling quantization noise and model adaptation), we’re curious if you’ve thought about how Shampoo’s performance might compare to other optimizers under these alternative schemes? We also wonder if testing a broader set of QAT methods could further validate whether Shampoo’s advantages are specific to QuEST or generalizable across common QAT paradigms.

4. Your scaling laws leverage Chinchilla’s full-precision optimal data-to-model ratio, which provides a consistent baseline for comparison. We note that quantization can alter a model’s information capacity, for example, 4-bit models may require more data to offset precision-related accuracy loss. We’re interested to hear your thoughts: how do you think reusing the full-precision data ratio might influence the assessment of Shampoo’s parameter efficiency? And would recalibrating the data ratio specifically for QAT be a valuable next step to confirm if Shampoo’s efficiency stems from inherent quantization robustness, rather than compatibility with full-precision settings?

5. You rightly point out that MMR and Kurtosis are limited by their focus on single-layer errors, which motivates the need for \(R_\ell\). We’re curious, have you explored the possibility of refining these traditional metrics (e.g., a weighted MMR that assigns different importance to attention vs. FFN layers) to better capture cross-layer error effects? If an improved MMR showed stronger correlation with quantization performance, how do you think it might complement (or compare to) \(R_\ell\), and could there be synergies between refining existing metrics and advancing new ones like yours?

6. Your conclusion thoughtfully highlights that optimizers performing well in full precision may not retain that advantage under QAT, a key insight for practical LLM deployment. We’re wondering if you’ve considered discussing the potential design direction of optimizers that excel in both full precision and QAT (e.g., quantization-adapted variants like a modified Lion optimizer)? We’d be interested to hear how such dual-excellence optimizers might align with your current findings, and whether you see this as a valuable direction for future work to build on your conclusions.

**Questions:**

In your full-precision training, you use the ClimbMix dataset (400B tokens) but do not analyze how data domain diversity (e.g., mixing technical vs. conversational text) interacts with optimizer-quantization performance. For instance, if a subset of ClimbMix has more outlier-prone features, might this disproportionately favor Shampoo? Did you conduct any ablation on dataset subsets to rule out data bias in your optimizer comparisons?

---

> ### Author Response · Authors · 2025-11-23
> **Reply to Review**
>
> Thanks for your time reviewing our paper.
>
> Replies to Weaknesses:
> 1) **Choice of model**: We chose the OLMo 2 backbone because it uses the latest architectural advances (e.g. QKNorm, GQA), serving as a strong and up-to-date baseline (see also https://allenai.org/blog/olmo3). Nevertheless, the effect of different optimizers on the linear layers should be similar (see Figure 4), and thus we would expect similar results for alternative architectures (including older architectures like GPT-2, Llama 1/2/3).
> To confirm this, we are currently rerunning our 350M experiments for Llama 2 on FineWeb. We will update this thread, as well as the main text, as soon as we obtain our results (probably in a few days).
> 2) **Complexity of calculating R**: This is an insightful comment. Calculating R can indeed be demanding, requiring four forward passes. However, we feel it is worth it, given how much more correlated it is to quantization degradation. As seen in Figure 1, we found that MMR/Kurtosis, albeit fast, do not really achieve their goal of correlating with quantization degradation. Unfortunately, due to the sequential nature of the computation (you need $R_{\ell-1}$ to calculate $R_\ell$), it is not really possible to isolate key layers. Nevertheless, we include an optimized implementation in our codebase, which closely follows the mathematical derivation of R. We will update the text accordingly.
> 3) **Choice of PTQ methods**: In terms of PTQ, we extended our study to also include GPTQ for the 500M models. The results, shown in the table below, closely follow the RTN findings. While extending our study to other QAT methods would indeed be reassuring of the generality of our results, it is unfortunately beyond our computational budget. However, given that all other results (PTQ, theoretical, QuEST) show the same trends, we expect Shampoo's advantage to still hold for other QAT schemes.
> |Optimizer|Avg. zero-shot, GPTQ (W4A4)|
> |---|---|
> |AdamW|55.47|
> |Muon|50.22|
> |PSGD|53.73|
> |Scion|52.21|
> |Shampoo|**55.84**|
> |SOAP|51.16|
> 4) **Data-to-model ratio**: Quantization can indeed reduce the information capacity/effective number of parameters, and thus indirectly decrease the resulting model-data ratio. To see what happens at such a reduced ratio, we are rerunning our QAT 350M experiments with a four-times lower amount of training data.
> 5) **Refining traditional metrics**: Unfortunately, this does not appear to be a promising direction: traditional metrics (e.g., MMR/Kurtosis), by their definition, only consider single layers (one cannot calculate MMR/Kurtosis across activations of different layers, since they come from different distributions). Taking the weighted average will not fix this inherent limitation, since it still cannot capture non-linear interactions across layers (like R does). Lastly, heuristics like weighted averages introduce new arbitrary choices (e.g., what should we set the weights to), complicating their practical applications.
> 6) **Designing optimizers that excel in full precision/QAT**: That is indeed an interesting future direction. We thank you for this suggestion, and will include a brief discussion in the paper. Many optimizers (e.g., Muon, PSGD) internally solve a constrained optimization problem. By adapting the problem to also include a condition about the gain G, so that the final weights propagate quantization error only mildly, one might indeed be able to obtain an optimizer that combines the best of both worlds. This is a significant contribution of our analytic study.
>
> Additional Questions:
> 1) We agree that we should confirm that we do not "overfit on ClimbMix", and thus are running our new Llama 2 experiments on a new dataset (FineWeb). An extensive analysis about how the dataset might influence optimizer/quantization performance would certainly be interesting, but unfortunately diverges from the main theme of the paper, which is about how optimizers affect quantization.

---

> > ### Author Response · Authors · 2025-11-27
> > **Reply Update**
> >
> > Dear Reviewer,
> >
> > We updated the manuscript with:
> > 1) Experiments on 350M Llama 2 models and FineWeb. The results are similar to the OLMo2 models:
> > |Optimizer|Avg. zero-shot, Llama 2 (W4A4)|
> > |---|---|
> > |AdamW|47.35|
> > |Muon|45.20|
> > |PSGD|48.01|
> > |Scion|47.77|
> > |Shampoo|**50.86**|
> > |SOAP|48.13|
> > 2) A discussion about the complexity of calculating R (see footnote in page 6)
> > 3) The GPTQ results (shown above)
> > 4) QAT experiments for a reduced data-to-model ratio of 5 (instead of the Chinchilla-optimal 20), and 350M models. The results are again similar to our original experiments. Shampoo shows the smallest quantization degradation, even though it does not lead to the best model at this scale (similarly to Table 4).
> > |Optimizer|Avg. zero-shot, D/N=5 (W4A4)|
> > |---|---|
> > |AdamW|51.30 (-3.69)|
> > |Muon|52.75 (-5.05)|
> > |PSGD|49.68 (-7.88)|
> > |Scion|53.23 (-2.71)|
> > |Shampoo|52.51 (**-2.54**)|
> > |SOAP|**53.45** (-2.66)|
> > 5) A brief discussion about how our analysis can be used for the design of a quantization-friendly optimizer (see conclusion)
> >
> > We also improved the paper with the suggestions of the other reviewers (a summary of all the changes can be found as an Official Comment).
> >
> > As the discussion is soon drawing to a close, we wanted to send you a gentle reminder regarding the discussion. We would be very happy if you could please examine our responses and determine whether they addressed your concerns.
> >
> > Best regards,
> >
> > The authors

---

### Author Response · Authors · 2025-11-27
**Manuscript update**

Dear Reviewers and AC,

Thanks again for taking the time to read our paper. We improved the manuscript with the suggestions of all the reviewers. Specifically, we included:
1) In the Appendix, experiments with 500M models and GPTQ, 350M Llama 2 models on FineWeb, and 350M QAT runs for a reduced data-to-model ratio of 5 (instead of 20). Reassuringly, we observe the same trends as in the original experiments. All these experiments together required about 1000 extra H100 hours (along with engineering efforts), and hence we could not update earlier.
2) A discussion about the complexity of calculating R (see footnote in page 6)
3) A brief discussion about how our analysis can be used for the design of a quantization-friendly optimizer (see Conclusion)
4) A citation to "Gpt3. int8 (): 8-bit matrix multiplication for transformers at scale." in the Introduction
5) Comments in the captions of Figures 3 and 4 about the oscillations
6) Various fixes suggested by @Reviewer 7Mqb (one version of Panferov et al. 2025, better caption for ABC decomposition of Figure 3, removed multiple definitions of MMR)
7) Assorted other fixes (typos, text improvements, visual improvements etc.)

We would love to know if we addressed your concerns, and how we could further improve our work.

Looking forward to talking with you,

The Authors

---

> ### Author Response · Authors · 2025-12-03
> **Post OpenReview bug comment**
>
> Dear Reviewers and AC,
>
> Unfortunately, although we tried to reply to the reviews as early as possible, we did not manage to engage in a discussion with the reviewers due to the OpenReview bug. The exception was @Reviewer Ag8F, who maintained his advocacy for our paper.
>
> We did our best effort to reply to all the reviewers' concerns as comprehensively as possible. Specifically, we run three sets of new experiments (500M GPTQ, 350M Llama 2, 350M QAT), which required about 1000 extra H100 hours.
>
> We do not repeat our contributions or our replies here, to avoid repeating information and thus facilitate the task of the AC. Our replies are in the same threads as the original reviews, and numbered in a similar way. We moreover included a general summary with all the improvements in the manuscript, in our immediately preceding post.
>
> We remain at your disposal,
>
> The Authors

---

### Meta-Review · Area_Chair_BbiT · 2025-12-31

**Summary:**

This paper discusses a valuable question: how does the choice of optimizer affect model performance in the presence of quantization? The paper investigates the impact of optimizer selection on model robustness under quantization, considering both post-trained quantization (PTQ) and quantization-aware training (QAT).

The paper was assigned to three reviewers. Two reviewers expressed a negative attitude (score: 4), and for various possible reasons, neither responded to the authors' rebuttal. One reviewer gave a higher score (score: 8) and maintained it. The authors' rebuttal indicate that they conducted extensive experiments to validate the paper's points and provided detailed rebuttal to the reviewers' questions. Although the authors did not receive any score improvements from the reviewers, we believe that these rebuttals further support the core arguments of the paper.

Based on the above, I recommend acceptance.

**Reviewer Concerns:**

Reviewer f8Mk： Extending these experiments to more mainstream LLM architectures like Llama or GPT-2 (or even Llama 3, which is widely used in industry)?  How does Shampoo's performance compare to other optimizers under other QAT variants? how do you think reusing the full-precision data ratio might influence the assessment of Shampoo’s parameter efficiency? **In their rebuttal, the authors conducted further experiments to verify the paper's points, and I believe these concerns will be resolved.**

Reviewer 7Mqb：Can the authors point to the specific literature that claims outliers as the problem? Can the authors point to the specific literature that claims outliers as the problem? The oscillations in Figure 3 and more pronounced in Figure 4 warrant some attention. What is going on in Fig. 4? Why are oscillating between regular values between successive layers? The experiments are across different number of parameters, but the results and observations in this paper might be overfitting to a single architecture family. What is the justification? And would the authors expect these observations to generalize to other architecture families? **The author provided a detailed response in the rebuttal, and I believe these issues will be resolved.**

Reviewer Ag8F：**The reviewers have replied that they have no further questions.**

**Reviewer Scores:**

Reviewer f8Mk:(score:4) The reviewer's score might be raised to 6 if they are fully involved in the discussion.

Reviewer 7Mqb:(score:4) The reviewer's score might be raised to 6 if they are fully involved in the discussion.

Reviewer Ag8F:(score:8)

---

### Decision · Program_Chairs · 2026-01-26

Accept (Poster)